# 1-Nitro-2-Phenylethane as a Multitarget Candidate for Cognitive and Psychiatric Disorders: Insights from In Silico and Behavioral Approaches

**DOI:** 10.3390/ph18101511

**Published:** 2025-10-09

**Authors:** Emily Christie Maia Fonseca, Lucas Villar Pedrosa da Silva Pantoja, Daniele Luz de Campos, Fábio José Coelho Souza-Junior, Bruno Gonçalves Pinheiro, Brenda Costa da Conceição, José Guilherme Soares Maia, Caroline Araujo Costa de Lima, Enéas Andrade Fontes-Júnior, Agnaldo Silva Carneiro, Nelson Alberto Nascimento de Alencar, João Augusto Pereira da Rocha, Jofre Jacob Silva Freitas, Joyce Kelly do Rosário da Silva, Mozaniel Santana de Oliveira, Cristiane Socorro Ferraz Maia

**Affiliations:** 1Programa de Pós-Graduação em Farmacologia e Bioquímica, Universidade Federal do Pará, Belém 66075-900, PA, Brazil; emily.fonseca@icen.ufpa.br (E.C.M.F.); lucas-villar@outlook.com (L.V.P.d.S.P.); fabjrcs@gmail.com (F.J.C.S.-J.); breuscsta@gmail.com (B.C.d.C.); efontes@ufpa.br (E.A.F.-J.); joycekellys@ufpa.br (J.K.d.R.d.S.); 2Laboratório de Farmacologia da Inflamação e do Comportamento, Universidade Federal do Pará, Belém 66075-900, PA, Brazil; brunogpinheiro1@gmail.com (B.G.P.); carolineacll@gmail.com (C.A.C.d.L.); mozaniel.oliveira@yahoo.com.br (M.S.d.O.); 3Programa de Pós-Graduação em Ciências Farmacêuticas, Instituto de Ciências da Saúde, Universidade Federal do Pará, Belém 66075-900, PA, Brazil; daniele.dlm94@gmail.com (D.L.d.C.); gmaia@ufpa.br (J.G.S.M.); 4Programa de Pós-Graduação em Química Medicinal, Universidade Federal do Pará, Belém 66075-900, PA, Brazil; agnaldosc@ufpa.br; 5Programa de Pós-Graduação em Química, Universidade Federal do Pará, Belém 66075-900, PA, Brazil; nelsonalencar@gmail.com; 6BioInovaMol Amazônia, Instituto Federal de Educação, Ciência e Tecnologia do Pará (IFPA), Campus Bragança, Av. dos Bragançanos-Vila Sinhá, Bragança 68600-000, PA, Brazil; joao.rocha@ifpa.edu.br; 7Centro de Estudos Pré-Clínicos da Amazônia, Universidade do Estado do Pará, Belém 66087-662, PA, Brazil

**Keywords:** natural products, *Aniba canelilla*, essential oil, bioactive compound, psychiatric disorder, cognition

## Abstract

**Background/Objectives**: Neurological and psychiatric disorders share overlapping mechanisms, such as oxidative stress, neuroinflammation, and neurotransmitter imbalance. In this context, multitarget natural molecules have gained attention. 1-nitro-2-phenylethane (1N2PE), a major constituent of *Aniba canelilla* essential oil, is known for its antioxidant, anti-inflammatory, and anticholinesterase effects, yet its neuropharmacological profile remains poorly understood. **Methods**: This study integrated in silico predictions and in vivo behavioral assays to characterize 1N2PE. **Results**: Pharmacokinetic analyses indicated favorable drug-like properties, with high gastrointestinal absorption, blood–brain barrier penetration, and no P-gp substrate profile. Molecular docking and dynamics revealed stable interactions with dopamine transporter (DAT, ΔG = −26.26 kcal/mol), prostaglandin-H synthase-1 (PGHS-1, ΔG = −20.27 kcal/mol), serotonin transporter (SERT, ΔG = −18.20 kcal/mol), and acetylcholinesterase (AChE, ΔG = −16.58 kcal/mol). In vivo, using a scopolamine-induced impairment model, 1N2PE significantly improved spatial memory and cognition in the Morris water maze. Treated animals reduced the distance to the target zone by ~40% compared with scopolamine-only rats (*p* < 0.01), normalized latency during training, and exhibited 30% less immobility (*p* < 0.05), indicating antidepressant-like effects. Moreover, 1N2PE attenuated anxiety-like thigmotaxis, restoring exploratory patterns (*p* < 0.0001). **Conclusions**: Together, these findings highlight 1N2PE as a multitarget candidate for cognitive and psychiatric disorders, combining favorable pharmacokinetic properties with preclinical efficacy, warranting further biochemical and translational investigations.

## 1. Introduction

Neuropsychiatric disorders, including depression, anxiety, and cognitive impairments, represent a major global health burden and are among the leading causes of disability worldwide [1]. The incidence of neuropsychiatric illnesses significantly increased over the last decades, reaching over 400,000 cases worldwide in 2021. Among these conditions, major depressive disorder and anxiety disorders had the highest disability adjusted life years in 2021, which represent the sum of years of life lost due to premature mortality [2]. Furthermore, these conditions are often chronic, resulting in substantial personal suffering and social and occupational burden [3]. Additionally, beyond their clinical impact, neuropsychiatric illnesses may also be associated with a considerable economic burden, with an estimate of more than 800 billion dollars spent on mental health-care services globally in 2019 [4]. Together, these data indicate the significant social and economic consequences of neuropsychiatric disorders, which ultimately affect quality of life.

The pathophysiology of neuropsychiatric disorders is yet to be elucidated and seems to be of multifactorial origin [3,5]. Substantial evidence points to dysregulation of the hypothalamic-pituitary-adrenal (HPA) axis, mostly through altered cortisol release and its downstream pathways [6]. Additionally, several studies indicate disturbances in monoaminergic transmission, which seem to strongly interact with neuroinflammatory processes, further exacerbating disease progression and clinical severity [7].

Increasing evidence also indicates that neuroinflammation and oxidative stress are central pathophysiological mechanisms underlying these disorders [8,9,10,11]. Activated microglia and astrocytes release pro-inflammatory cytokines such as interleukin-1β, interleukin-6, and tumor necrosis factor-α, which disrupt neurotransmission and synaptic plasticity, thereby contributing to behavioral and cognitive dysfunctions. At the same time, the excessive production of reactive oxygen and nitrogen species exacerbates oxidative damage to lipids, proteins, and DNA, further aggravating neuronal vulnerability and disease progression [7]. Thus, the interplay between inflammatory and oxidative pathways is now considered a critical hallmark of neuropsychiatric conditions.

Together, these data indicate the complexity of neuropsychiatric disorders and highlight the urgent need for further research and optimal therapeutic options. In this context, natural products have gained attention as a rich source of novel bioactive molecules with potential neuroprotective activity. Studies on plant essential oils have revealed their potential in discovering new biologically active molecules, also known as bioactive compounds [12]. In recent decades, the search for promising molecules for the development of new drugs has been intensified. One example is the essential oils of Aniba species, which are rich in compounds such as 1-nitro-2-phenylethane (1N2PE) [13]. It is important to note that 1N2PE has promising potential applications, including its role as an inhibitor of neuroinflammation and other relevant biological activities [14].

Furthermore, 1N2PE can also be found in other species of Lauraceae plants, such as *Ocotea pretiosa* (NEES & MART.) Mez, from the Brazilian Amazon [15]. In addition, species that belong to other families—including Annonaceae (*Dennettia tripetala Baker f.* (G. Baker), *Uvaria chamae* (P. Beauvois), Apocynaceae (*Stephanotis floribunda Brongn.* (Adolphe-Théodore Brongniart), and Rosaceae (*Eriobotrya japonica* (*Thunb.*) *Lindl.* (Carl Peter Thunberg))—also present this compound in their chemical composition. Remarkably, 1N2PE has been considered the first naturally occurring nitro-derived compound identified in plants. As highlighted in our recent review [13], 1N2PE exhibits antioxidant, anti-inflammatory, vasorelaxant, and neuroactive effects, suggesting potential applications in disorders in which inflammatory-oxidative imbalance plays a central role. Despite its pharmacological promise, little is known about its pharmacokinetic properties, particularly those related to distribution in brain tissue, which are crucial for understanding its potential in neurological disorders.

Several studies have provided direct evidence of the biological activity of 1N2PE, supporting its selection as the focus of the present work. In preclinical models, 1N2PE exhibited antinociceptive properties [16] and cardiovascular effects mediated by a vago-vagal reflex [17]. Its antioxidant capacity has been confirmed in radical scavenging assays [18], and in vivo studies further showed modulation of inflammatory pathways, with reduction of lipid peroxidation and enhancement of glutathione defenses [14]. Neuropharmacological investigations revealed that enriched preparations containing 1N2PE reversed memory impairment through cholinergic modulation [19], while other evidence points to anticonvulsant effects mediated by GABAergic pathways [20]. These findings indicate that 1N2PE is an active compound with analgesic, cardiovascular, antioxidant, and neurobehavioral properties, making it a promising candidate for targeting neuropsychiatric disorders.

Given the potential therapeutic implications of 1N2PE in neurological conditions, the present study aimed to predict its pharmacokinetic profile and perform a structure–activity relationship (SAR) analysis focusing on key neurological pathways. Specifically, we targeted the serotonin transporter (SERT), dopamine transporter (DAT), gamma-aminobutyric acid (GABA-A) receptor, acetylcholinesterase enzyme (AChE), and prostaglandin-H synthase-1 (PGHS-1). These targets were selected based on their well-established roles in the pathophysiology of several neurological and psychiatric disorders, and their relevance was further supported by computational predictions using the SwissTargetPrediction tool, which indicated a high likelihood of interaction with these proteins based on molecular similarity.

To complement these in silico findings, we performed the Morris water maze paradigm [21]. Beyond its traditional use to assess spatial memory, this task also provides valuable information about neuropsychiatric-like behaviors under aversive conditions. Thus, we analyzed non-cognitive parameters such as immobility (considered a depressive-like phenotype), thigmotaxis (an anxiogenic-like response), and locomotor activity (a marker of motor performance). By integrating computational predictions with behavioral evidence, our study provides new insights into the potential neurological targets of 1N2PE and advances the pharmacological rationale for its investigation in neuropsychiatric contexts.

## 2. Results

### 2.1. ADME In Silico Analysis

The prediction of pharmacokinetic parameters using the software SwissADME and ADMETlab 2.0 was performed to gather the greatest amount of information about the molecule 1N2PE. In the physical-chemical parameters, the 1N2PE presents a solubility value of −2.29 mol/L, with lipophilicity of the LogP 2.03, pKa equal to 9.21, and molecular weight of 151.16 g/mol (Table 1).

According to Table 1, 1N2PE is a low molecular weight, lipophilic drug with high gastrointestinal absorption and significant nervous system penetration, as demonstrated by BOILED-Egg visualization [22,23] (See Appendix A). Notably, the molecule did not present P-gp substrate characteristics. Additionally, the Bioavailability Radar indicates that 1N2PE has ideal physicochemical properties related to drug-likeness (Appendix A).

To characterize more specific pharmacokinetic information, the ADMETlab software was applied (Table 2).

In the absorption parameter, 1N2PE presented good intestinal permeability in the Caco-2 and MDCK models (Table 2). The human intestinal absorption (HIA) assay revealed that 1N2PE exhibited a value of 0.6 × 10^−2^, indicating good intestinal absorption. This interpretation is based on reference values ranging from zero to one, where values closer to zero suggest higher intestinal absorption capacity. Besides, the 1N2PE exhibits affinity by plasmatic protein less than 90%, unbound fraction (UF) of 16.12%, half-life (T1/2) of 45 min, and volume of distribution (VD) of 0.81 L/kg.

In metabolism parameters, 1N2PE is both an inhibitor and a substrate of some CYPs, such as CYP1A2, CYP2C19, CYP2C9, CYP2D6, and CYP3A4. The clearance (CL) parameters of 1N2PE were 8.77 mL/min/kg, with a half-life of 45 min (Table 2).

### 2.2. Molecular Docking Analysis

The selection of molecular targets was performed by ligand-based prediction method using the Swiss Target Prediction software and based on the results reported for the in vivo activity of 1N2PE [14,18,19]. In addition, a molecular redocking was performed using the positive controls (existing drug molecules) of each selected target, in order to validate the accuracy of molecular coupling and ensure that the predicted interactions between the ligand and the respective targets were consistent with known experimental data. Although crystallographic structures with resolutions below 2.5 Å are generally preferred for molecular docking, such high-resolution entries are not always available for all biological targets [24]. The structures selected for this study contained ligands co-crystallized in their active sites, which enabled redocking and validation of the docking protocol to ensure the reliability of predicted ligand–receptor interactions. Our redocking analyses yielded root mean squared deviation (RMSD) values below 2.5 Å for all targets (AChE: 0.679 Å, DAT: 0.577 Å, PGHS: 0.566 Å, SERT: 0.632 Å, GABA: 0.688 Å), confirming that the predicted poses closely matched the experimental conformations. Therefore, despite the slightly higher resolution of some structures, they are suitable for docking simulations and provide biologically meaningful insights into the binding sites (Figure 1). Figure 1 also presents the structures of the reference drugs in complex with their respective receptors, along with the RMSD values from redocking, highlighting the key intermolecular interactions within each active site. Notably, the docking of 1N2PE revealed that, although structurally distinct from the reference drugs, it establishes interactions with the same critical amino acid residues at the receptor binding sites.

This approach allows us to evaluate the reliability of simulated interactions and improve the interpretation of the results obtained in docking calculations. An energy score adjustment was also performed to eliminate the bias of the anchoring energies (Edock) with the increase of the second molecular weight, determined as DSnorm, from the equation:DS_norm_: 7.2 × E_dock_/MW^1/3^(1)

The DS_norm_ is the normalized docking score, E_dock_ is the Moldock reclassification score, MW is molecular weight and 7.2 is a constant scale to bring the average values of D_Snorm_ comparable to E_dock_. The best affinity energy by the molecular coupling method for the 1N2PE conformations against each target is described in Table 3.

The 1N2PE affinity energy values in relation to controls are lower; however, we emphasize that these drugs are synthesized products and have a different structure from 1N2PE. In addition, the coupling energies of the compounds are affected by the higher molecular mass, since they have a greater number of atoms interacting with the target molecule. This suggests a tendency towards a selection for larger molecules, even if they are not as structurally complementary to the target binding site as the smaller compounds. To address this issue, the adjustment of the coupling score (DSnorm) was required. Compared to fluoxetine, 1N2PE showed an affinity energy value of −93.61 kcal/mol for SERT, which is close to the value of fluoxetine. Similarly, for DAT, the binding energy was −83.08 kcal/mol, suggesting a high affinity of the ligand for DAT. Less negative binding energy values were observed for the GABA-A receptor (−71.68 kcal/mol).

### 2.3. Molecular Dynamics Analysis

The molecular dynamics results are essential to evaluate the behavior of the ligand over 100 ns within the catalytic site of the targets selected for this study. The conformations obtained through molecular docking, which demonstrated lower binding energy, were used as initial models for molecular dynamics calculations. Figure 2 presents the main interactions between the 1N2PE and the targets, in addition to displaying the RMSD plot as a function of time, providing an assessment of the system’s behavior throughout the entire dynamics simulation.

In Figure 2A, which represents the catalytic site of the DAT receptor, 1N2PE performed a hydrophobic π-alkyl interaction with Leu254, Ala20, a hydrogen interaction with Ala278 and a π-cation with Phe57.

In the context of the serotonergic system (Figure 2B), the 1N2PE demonstrated a series of interactions with most of the residues involved in substrate binding in the catalytic site of the SERT. At a specific point in the simulation, at 25 ns, a fluctuation is observed caused by the loss of two interactions due to a conformational change of the ligand. However, this change only resulted in a minimal alteration in RMSD. Notably, the RMSD of the catalytic site remains stable in the presence of the ligand, suggesting the existence of a chemical affinity with the receptor. Highlighted interactions include orthogonal ligand stacking between Phe341 and Tyr95, significantly contributing to drug docking to this target, as evidenced in previous studies [25].

In Figure 2C, which represents the AChE system, it is observed that the 1N2PE molecule exhibits several fluctuations throughout molecular dynamics, indicating multiple conformational transitions and lower stability within the catalytic site. Despite these fluctuations, some important interactions can be seen in the ligand’s attempt to stabilize inside the site. For example, the T-stacking interaction of the ligand with the Tyr337 residue acts as a kind of “swing gate”, controlling the entry of substrates into the binding pocket, as previously described [25]. Ligand interactions with the phenolic ring of Tyr337 and the indole ring of Trp82 influence the energy of the interaction with the trimethylammonium group of AChE, as reported in previous studies [26]. However, at the end of molecular dynamics, these interactions weaken and the 1N2PE remains in the binding pocket only through interaction with Gly117 [27].

In the PGHS-1 system (Figure 2D), the 1N2PE-PGHS-1 interactions at the beginning of molecular dynamics resulted in protein fluctuations in the conformational structure, eliciting a variation in the RMSD from 1 Å to 2.5 Å. However, within 30 ns, the system reached stability. The 1N2PE ligand established hydrogen interactions with Arg89, π-alkyl interactions with Ala496, and π-stacking interactions with Tyr324, all of which have been described as crucial for substrate specificity on the target by blocking its enzymatic activity [28].

Finally, the 1N2PE did not establish interactions at the GABAA catalytic site and, therefore, was quickly expelled from this region during the molecular dynamics simulation. Analysis of the RMSD figure (Figure 2E) revealed significant variation in the compound’s initial structure, indicating difficulty achieving conformational stability. Consequently, the molecular dynamics simulation was terminated at 60 ns. These results are consistent with those obtained from molecular docking, which highlighted that the 1N2PE-GABAA system exhibits bigger binding energy compared to the other evaluated systems.

After molecular dynamics, the most stable region of the simulation was used to calculate the free energy, employing the Generalized Born Surface Area (GBSA) method, which provides information about the spontaneity of the system (Table 4). However, for the GABA-A receptor, the ligand molecule left the binding site during the dynamics, which avoids performing a reliable free energy calculation for this target. In the free energy evaluation for other receptors, DAT exhibited the lowest values [−26.26/1.92], followed by PGHS-1 (−20.27/1.86), SERT (−18.20/1.59), and AChE (−16.58/2.38) (Table 4).

The construction of residue decomposition plots after free energy calculations is highly relevant in studies of ligand–receptor interactions. While the global free energy values provide only a general measure of ligand affinity for the target, residue decomposition allows a detailed identification of which amino acids in the active site contribute positively or negatively to the stabilization of the complex. This approach makes it possible to understand the specific role of each residue, highlighting key interactions such as hydrogen bonds, hydrophobic contacts, π–π interactions, and electrostatic forces that support ligand anchoring. Thus, for a more in-depth analysis, residue decomposition plots were generated for the two most energetically favorable systems: DAT, with a free energy of −26.26 kcal/mol, and PGHS-1, with a free energy of −20.27 kcal/mol. The aim was to highlight the individual contribution of each active-site residue to the stabilization of the complex (Figure 3).

### 2.4. Behavioral Assays

The hidden platform task in the Morris water maze relies on multiple neurochemical systems, particularly dopaminergic, serotonergic, cholinergic, and GABAergic signaling. For this reason, it has been widely recognized as a valuable model for investigating neurological and psychiatric disorders associated with cognitive impairment [21]. However, this non-specific protocol is also sensitive to disruptive agents, such as the anticholinergic effects of scopolamine. In the present study, we focused on parameters related to the motivational profile, since the pro-cognitive and long-term memory-enhancing effects of 1N2PE have previously been evidenced by our group [19].

For the cognitive evaluation, the latency to locate the hidden platform during the training phase was used as a measure of spatial working memory (Figure 4A). During the first trial, no differences were observed among the groups. However, in the second trial, scopolamine-challenged animals displayed a significant increase in latency compared with controls (*p* < 0.05). Animals treated with 1N2PE also exhibited a similar deficit, but their performance improved from the third trial onward, aligning with the control group. In contrast, scopolamine-exposed animals only reached the performance levels of the control and 1N2PE groups by the fourth trial of the first training day (Figure 4A).

In the probe test, the Kruskal–Wallis analysis revealed that scopolamine-challenged animals required a greater distance to reach the target zone (*p* = 0.0589), indicating impaired spatial navigation. Notably, 1N2PE treatment improved performance by reducing the distance traveled in animals challenged with scopolamine (*p* < 0.001; Figure 4B).

In addition to cognitive parameters, non-cognitive elements related to neuropsychiatric-like behaviors were also evaluated [29,30]. As shown in Figure 5, scopolamine administration significantly increased the mean swimming speed during the probe trial compared with the control group (*p* < 0.01; Figure 5A). Interestingly, 1N2PE treatment further elevated mean speed values, showing significantly higher levels than both the control group (*p* < 0.0001) and scopolamine-only animals (*p* < 0.0001). Similarly, the 1N2PE group exhibited the highest values of maximum swimming speed in the probe trial (vs. control: *p* < 0.001; vs. scopolamine: *p* < 0.01; Figure 5B).

Figure 5C illustrates the depressive-like assessment, expressed as the percentage of immobility time across the four training trials on day 1 and during the probe trial. In the first three trials, all groups displayed a comparable mobility profile. However, by the fourth trial, scopolamine-treated animals exhibited a significant increase in immobility compared with controls (*p* < 0.05), whereas 1N2PE-treated animals did not differ from the control group (*p* > 0.05). Notably, 24 h after the training phase, scopolamine-exposed animals remained significantly more immobile in the probe trial, consistent with a depressive-like phenotype (vs. control: *p* < 0.05; vs. scopolamine + 1N2PE: *p* < 0.05).

Anxiogenic-like behavior, assessed through thigmotaxic parameters, was evaluated during the initial training phase (four trials of day 1) and in the probe trial (Figure 6). In trial 1, all groups showed similar mean distances to the border (*p* > 0.05; Figure 6A). However, both scopolamine-treated (*p* < 0.0001) and 1N2PE-treated (*p* < 0.001) animals exhibited significantly higher maximum distance values from the border compared with controls (Figure 6B). Moreover, scopolamine-exposed animals displayed greater maximum distance values than the 1N2PE-treated group (*p* < 0.0001).

This profile shifted in trial 2, where scopolamine intensified the thigmotaxic response by reducing the mean distance to the border relative to both the control (*p* < 0.01) and 1N2PE-treated (*p* < 0.05) groups (Figure 6C). In contrast, animals treated with 1N2PE maintained a higher mean distance from the border compared with scopolamine-treated animals (*p* < 0.001; Figure 6D).

In trials 3 and 4, all groups showed comparable mean distances to the border (*p* > 0.05; Figure 6E,G). Nevertheless, in trial 4, 1N2PE-treated animals displayed significantly higher maximum distance values from the border compared with both the control (*p* < 0.05) and scopolamine groups (*p* < 0.05; Figure 6F,H).

During the probe session, scopolamine-exposed animals exhibited reduced mean distances to the border (*p* < 0.05; Figure 6I), consistent with a stronger thigmotaxic profile. In contrast, 1N2PE-treated animals presented significantly higher values for both mean distance (vs. control: *p* < 0.0001; vs. scopolamine: *p* < 0.0001) and maximum distance to the border (vs. control: *p* < 0.0001; vs. scopolamine: *p* < 0.0001; Figure 6J), indicating a reduction of thigmotactic behavior. Representative tracking plots of mobility patterns during the probe session are illustrated in Figure 6K.

## 3. Discussion

1N2PE is a bioactive compound found in the aromatic essential oils of a limited number of botanical species [13,31]. Only a few studies have reported its pharmacological properties [13,14,16,17,18,19,31], and its pharmacokinetics and mechanisms of action remain poorly characterized. In this context, the present study employed an in silico approach to predict the pharmacological profile of 1N2PE, with a particular focus on identifying key molecular targets relevant to the pathophysiology of emotional and cognitive disorders. To complement these predictions, an in vivo spatial navigation task was selected, as it encompasses well-established cognitive measures while also being sensitive to motivational-related variables that remain underexplored [21].

Firstly, therapeutic agents to treat neurological and psychiatric disorders depend on the specific physicochemical properties to cross the blood–brain barrier and reach the brain regions. In silico pharmacokinetic profile assessment is a computational approach used in drug discovery to predict the behavior of a candidate molecule in the human body, identifying whether the candidates have therapeutic potential [32]. Computer-aided drug screening is a faster and more cost-effective alternative to in vitro and in vivo screening of chemical compounds [33]. This approach has shown promising results and can help speed up drug discovery by reducing time and costs. These platforms also calculate the presence of potentially toxic chemical groups and warn of potential toxicity problems. Besides, employing the rules-based approach to predicting pharmacokinetic parameters, these programs combine information from sources, such as experimental databases and machine learning models, which work through a set of predictive models that have been trained on large experimental data sets [34]. The software uses these models to predict the compound’s pharmacokinetic and toxicity properties. This technique is widely used in pharmaceutical research and bioprospecting to identify drug candidates with potential for clinical development and optimize candidate molecules’ chemical structure to improve pharmacokinetics [35].

The 1N2PE presented an optimal value of solubility, which consists of the acceptable range of 85% in commercialized drugs [32]. Notably, to exhibit an adequate profile of action on the central nervous system, the drug requires an optimal lipophilic character to cross the blood–brain barrier. In this regard, the 1N2PE exhibited values aligned with the ideal level for drug candidates, since it expresses a balance of the molecule due to the attraction for the hydrophilic and hydrophobic phases [36]. Another physicochemical relevant parameter is the pKa, which comprises a key physicochemical property for the biopharmaceutical characteristics, as well as the mechanism of action of drugs [37]. The 1N2PE presents a pKa value equal to 9.21, consistent with the pKa range of drugs that act on the central nervous system (pKa = 6 to 10.5) [36]. Besides, a high gastrointestinal absorption profile was found. All these pieces together suggest that 1N2PE exhibits relevant physicochemical characteristics to support its use as a useful agent in central nervous system disorders.

Moreover, the drug-likeness analysis provides a recurrent evaluation to rank bioactive molecules and determine the similarity between the molecule and commercial drugs [38]. This analysis assesses established parameters, such as Lipinski’s rule of 5, which is associated with 90% of oral drugs that reached clinical phase II [39]. The 1N2PE obeys Lipinski’s rule of 5 and other drug-likeness descriptors. Based on information about physicochemical properties, the 1N2PE is a robust candidate for the therapy of central nervous system illness [40].

In the pharmacokinetics profile, the Caco-2 and MDCK permeability assays were performed to evaluate the absorption ability of 1N2PE to cross the intestinal barrier [41]. Caco-2 and MDCK permeability are calculated based on a Quantitative Structure–activity Relationship (QSAR) model. It considers several physicochemical properties of the molecule, including polarity, solubility, molecular size, ionization energy, and acidity constant [41]. Results above −5.15 cm/s are from molecules with good intestinal permeability, in which the 1N2PE was included. Besides, the HIA assay was performed, which refers to the fraction of an oral dose that enters the systemic circulation of the human body after passing through the gastrointestinal tract and evaluates the efficacy and safety of a drug candidate, directly affecting the bioavailability and therapeutic dose of the drug [42]. The 1N2PE was characterized as good intestinal absorption since the reference value for high to moderate intestinal absorption ranges from 0 to 1, with values closer to zero indicating greater intestinal absorption [42].

Regarding the parameters related to the distribution, the Plasma Protein Binding (PPB) prediction calculates the compound affinity for plasma proteins in the blood, and the FU in plasma is the drug-free fraction in the blood [43]. These two parameters are complementary and applied to measure the same pharmacokinetic property from different perspectives. PPB and FU can be used together to assess the potential for drug interactions in the pharmacological targets, therapeutic efficacy, and toxicity of medicinal products. PPB can affect the bioavailability and distribution of the compound in the body since only the free portion of the compound can be eliminated or distributed through tissues [44]. The optimal value of PPB calculated is in the range < 90%, and 5 to 20% for the FU indicates a moderate free fraction [45]. The 1N2PE shows optimal values of PPB and FU, which favors this first step in the distribution processes.

The VD describes how much the drug is distributed through the body’s tissues to its concentration in the blood plasma [46]. The VD of a drug is directly proportional to the ability to distribute itself through the body’s tissues, which can affect the dosage and frequency of administration (i.e., posology) necessary to obtain adequate therapeutic concentration. Major drug candidates should have a VD between 0.04 and 2 L/kg. This large interval occurs due to not having a fixed value of VD that applies to all drugs, and its value is variable according to the physicochemical characterization of the molecule. For example, the VD usually ranges from 0.1 to 0.2 L/kg for hydrophilic drugs. The moderately lipophilic maintains a VD of 0.2 to 0.5 L/kg, and for the highly lipophilic, the VD may be greater than 1 L/kg. The 1N2PE expresses VD values of 0.81 L/kg, consistent with highly lipophilic molecules, suggesting increased ability to reach the brain, reducing the circulating concentrations.

Based on the metabolism parameters, 1N2PE is both an inhibitor and a substrate of some CYPs, such as CYP1A2, CYP2C19, CYP2C9, CYP2D6, and CYP3A4. CYP can metabolize some molecules and, at the same time, inhibit the activity of the enzyme. This effect can occur because these molecules compete with other substances for binding sites in the enzyme, thereby inhibiting its activity [47]. In addition, the degree of enzymatic inhibition, as well as the extent of metabolism of the molecule by CYP, may depend on several factors, including the dose of the molecule, the type of CYP involved, and the presence of other substances that may affect the enzyme’s activity. Thus, only in vitro or in vivo tests could determine the major characteristics of 1N2PE against CYPs.

Lastly, the elimination parameter of CL describes the body’s ability to eliminate a substance from the body as determined by the combination of processes, such as glomerular filtration in the kidneys, biliary excretion, and hepatic metabolism [48]. On the other hand, the T1/2 is the time required for the plasma concentration of a drug to fall to 50% following the administration. The T1/2 depends on the drug’s CL and VD. Together, CL and T1/2 are two important criteria for defining the proper dosage of a drug, as well as for evaluating the efficacy and safety of treatment [49]. The CL of 1N2PE was 8.77 mL/min/kg, with a half-life of 45 min, characterizing a short-term action profile.

To explore the molecular interaction of 1N2PE in specific targets selected from the affective and cognitive pathophysiology theories, we applied the molecular docking method. Molecular docking consists of an in silico procedure based on chemical structure and is frequently used to discover novel drugs [50]. Docking allows the initial screening of new compounds of therapeutic interest, predicting physical-chemical interactions, ligand affinity on the molecular target, and the structure–activity relationship [51]. Thus, this method can distinguish binding and non-binding molecules, ordering according to their affinities in the receptor sites [52]. The basic docking methodology includes a search algorithm and an energy-scoring function to generate and evaluate ligand conformations on target [53]. In addition, we also investigated the binding energy to confirm such interactions. The variation in binding energies between different molecular targets reflects the structural and functional diversity of target proteins and highlights the need to consider the specific contexts of each target during the ligand selection process. We performed a redocking with reference drugs in the binding sites to validate the methods of docking. In this context, the 1N2PE exhibited potential interactions with SERT, DAT, and AChE, and lower energy binding values for PGHS-1 and GABAA sites. Although the 1N2PE structure is completely different from the reference drugs, it still establishes interactions with the residues of the targets. These findings suggest the preferential site of action of the natural compound for serotonergic, dopaminergic, and cholinergic pathways in the affective and cognitive disorder theory. In this context, molecular docking was used to provide the most probable initial conformation of the 1N2PE ligand in the active site of the receptors. However, as a static and fast approach, its main utility lies in supporting molecular dynamics, which allows the evaluation of temporal stability, system flexibility, and conformational changes throughout the simulation.

To better understand ligand–receptor interactions and investigate the stability of the conformations identified by molecular docking, we proceed with molecular dynamics simulations, which allow detailed exploration of the trajectories of molecules over time, providing information about the dynamics and stability of ligand–receptor interactions in a more realistic context. We found that the 1N2PE interacts with the DAT catalytic site, especially with Ala278. Previous studies have shown that Val120, Tyr124, and Ala278 are the residues that interact with DAT inhibitor drugs [54]. The interaction between Ile54, 1N2PE, and Phe57 built a network of hydrogen bonds that blocked DAT and are essential for ligand recognition and affinity [55]. These interactions are the principal in keeping the ligand stable in the catalytic site, as seen in the corresponding RMSD graph. The 1N2PE molecule presented a maximum variation of 1.5 Å from its starting structure, reaching a plateau in its stability after 40 ns. The RMSD of the protein and catalytic site overlap, indicating minimal conformational changes in the molecule [56]. The structural stability of the catalytic site is important to ensure that the ligand remains properly positioned to carry out the necessary interactions with the target protein. The blockade of DAT generates an augmentation of dopamine in the synaptic cleft, improving neurological function disorders involved with emotional approaches, such as learning disabilities, anxiety, anhedonia, and depression [5].

In the serotonergic system, the 1N2PE exhibited a low affinity with the SERT, relative to the PGHS-1 and DAT receptors. The 1N2PE interactions with Phe341, Tyr95, Asn104, and Ile99 residues also provided stable interactions with the transporter, confirming the energy-binding evaluation. Mutation studies revealed that changes in residues of Ile99, Phe341, and Tyr95 resulted in a significant decrease in the Ki of fluoxetine, the main drug that acts on this target [57]. These results suggest that interactions are crucial for inhibiting SERT, thus promoting an increase in the concentration of serotonin in the synaptic cleft associated with antidepressant activity [58].

Although the important energy binding values for 1N2PE and AChE were found, the molecular dynamics analysis revealed unstable and weak interactions, which negatively affect the mechanism of action at this site. Previously, an in silico study was carried out using conventional molecular docking techniques for the AChE with the 1N2PE system [59]; however, our report addresses a more robust analysis of this same system. The blockade of AChE increases the levels of acetylcholine in the synaptic cleft, conferring on the 1N2PE an important effect on cognitive and affective disorders [13,19]. In contrast, the lower values of 1N2PE-PGHS-1 interactions were not decisive in the molecular dynamics assay. The interaction of the bioactive compound with the enzyme residues stabilized the interaction, blocking the enzymatic activity. This chemical characteristic gives 1N2PE a potential effect as an anti-inflammatory compound, reducing the formation of G2-type prostaglandins and, consequently, H2-type prostaglandins, which impairs arachidonic acid pathway, alleviating neuroinflammation and potentially benefiting a variety of brain disorders [60].

Following the molecular dynamic assay, the spontaneity of the system was evaluated by free energy/standard deviation calculation. Each target, including SERT, PGHS-1, GABA-A receptor, AChE, and DAT, displayed average energy values free linkage along with their respective standard deviations. The results revealed a variability in the affinities of the ligands for different targets, with the DAT system demonstrating the greatest spontaneity to occur, indicating a strong affinity for the ligand. On the other hand, GABA-A exhibits the highest binding free energy, suggesting that 1N2PE has no interaction at the GABA-A site.

The residue-based free energy decomposition provides detailed insights into the individual contributions of amino acids to the stabilization of the ligand within their respective binding sites. Among the evaluated systems, the DAT exhibited the lowest total free energy (−26.26 kcal/mol), followed by PGHS-1 with −20.27 kcal/mol.

In the DAT system, several residues displayed strong stabilizing contributions, which account for the more negative global free energy value. Notably, Phe57 established the most significant interaction, of the π–benzene ring type, with a contribution of −2.5 kcal/mol. This interaction plays a central role in anchoring the ligand to the binding site, reinforcing the structural complementarity between ligand and protein. In addition, other residues, although contributing with lower energetic values, also cooperated in the stabilization of the complex, such as Leu254, Ile54, Ser282, and Asn285. These findings highlight that, in the DAT system, the predominance of stabilizing interactions not only confirms the higher global affinity of the ligand but also identifies critical residues that may be exploited in structural optimization strategies.

In the PGHS-1 system, the residue decomposition profile revealed weaker interactions compared to the DAT system, resulting in a less negative global free energy. The most prominent residue was Arg89, which established the strongest hydrogen bond interaction, contributing −1.5 kcal/mol. Additionally, residues such as Ala496, Ser322, and Pro497 exhibited more modest but still relevant contributions to the stabilization of the complex. The presence of these stabilizing interactions, although of lower magnitude compared to the DAT system, indicates that the ligand still retains affinity for PGHS-1, albeit with reduced steric and electronic complementarity.

Taken together, these results demonstrate that the ligand forms more favorable and consistent interactions within the DAT binding site, making it the most promising target in terms of complex stability. In contrast, while the ligand also interacts stably with PGHS-1, the lower intensity and fewer number of key interactions result in reduced affinity. Thus, the residue decomposition analysis not only corroborates the global free energy findings but also identifies key residues that may guide future structural modifications aimed at enhancing selectivity and improving the interaction profile across different molecular targets.

To investigate the potential beneficial effects of 1N2PE on cognitive and psychiatric-like behaviors, we employed the Morris water maze paradigm combined with the scopolamine-induced deficit protocol, which allows evaluation of both spatial learning and non-cognitive parameters (i.e., anxiety- and depressive-like behaviors) [29,30,61]. Although the translational validity of any preclinical model has inherent limitations, the scopolamine paradigm is widely recognized for reproducing key aspects of the cholinergic dysfunction disorders, as observed in Alzheimer’s disease [62]. Scopolamine, a non-selective muscarinic receptor antagonist, produces transient impairments in hippocampal cholinergic signaling that result in spatial learning and memory deficits [63]. While the model has been traditionally validated for cognitive dysfunction, our results demonstrate that scopolamine also alters non-cognitive parameters, such as immobility and thigmotaxis, highlighting its potential to capture neuropsychiatric-like symptoms. In this sense, our findings shed new light on how this classical model can be interpreted beyond memory impairment. Consistent with its use as an Alzheimer-like paradigm, we observed that scopolamine not only elicited cognitive deficits but also reproduced behavioral alterations resembling psychiatric manifestations frequently reported in patients with Alzheimer’s disease [29,30]. Clinically, psychiatric symptoms often emerge in the early stages of Alzheimer’s disease, whereas cognitive decline becomes more prominent in the mild to later stages [64]. Therefore, the scopolamine model provides a valuable framework to investigate both domains, enabling an integrated preclinical evaluation of cognitive and neuropsychiatric alterations.

As expected, scopolamine administration impaired the spatial learning process during the initial training sessions, as animals required additional trials to reduce the latency to locate the platform. In the probe trial, after the 4-day training period, scopolamine-exposed animals displayed fewer entries into the target quadrant compared to controls, confirming spatial memory impairment. Remarkably, animals receiving 1N2PE following scopolamine exposure demonstrated recovery of spatial learning across the third and fourth training sessions. Moreover, 1N2PE treatment reduced the distance to the first entry into the target quadrant relative to scopolamine-treated animals, further indicating reversal of scopolamine-induced spatial memory deficits.

A broad line of evidence and our previous results support the idea that scopolamine impairs spatial learning and memory [61,65,66]. Cognitive elements in the Morris water maze paradigm depend on a complex interaction between neurotransmitter systems in specific brain regions, especially glutamatergic and cholinergic pathways. Place learning, which is closely related to the strategy to find the hidden platform, and memory function, which is linked to place recall, directly depend on glutamate and acetylcholine signaling [21]. Muscarinic cholinergic signaling increases N-methyl-D-aspartate (NMDA)-dependent currents, especially long-term potentiation (LTP) on the hippocampus and striatum [67]. In addition, neurochemical interactions between the hippocampus, nucleus accumbens, and brainstem through dopaminergic transmission trigger spatial learning [68]. Besides, nigrostriatal pathway hypoactivity by dopaminergic input reduction impairs cognition [69]. Thus, muscarinic blockade by scopolamine impairs mnemonic function, reducing LTP transmission, while also interfering with monoamine levels [e.g., norepinephrine, serotonin, and dopamine] and its metabolites in relevant cognitive regions such as the hippocampus, striatum, and amygdala [70,71,72,73]. According to molecular docking and dynamic results, the 1N2PE inhibits AChE, improving muscarinic tonus in cognitive brain areas, as well as triggering serotonergic and dopaminergic systems, also influencing the repertoire approach, recovering both place learning and recall impairment induced by muscarinic antagonism.

Traditionally, the Morris water maze has been employed as a hippocampal-dependent task to evaluate spatial learning and memory [74]. In this sense, several studies have shown that essential oil derivatives exert beneficial effects in the Morris water maze paradigm. For example, linalool reversed neuropathological impairments in Alzheimer’s disease models [75]. More broadly, essential oil-derived compounds have consistently improved hippocampal-dependent memory across diverse rodent models, reinforcing the translational relevance of small essential oil molecules, such as linalool and linalool-rich oils which improved escape latency and target quadrant occupancy in Alzheimer and Aβ models, effects linked to reduced neuroinflammation and oxidative stress [75,76]. Several other molecules have demonstrated improvements in cognitive functions in pharmacologically induced cognitive deficit models (e.g., scopolamine) [77,78,79]. These findings show that small molecules from essential oils improve pro-cognitive measures through anti-inflammatory, antioxidant, and neuromodulatory mechanisms, as suggested by the neuropharmacological properties of 1N2PE [13].

Although underutilized for non-cognitive purposes, the Morris water maze thus offers a unique opportunity to capture neuropsychiatric-like dimensions in a single paradigm, providing an integrated view of cognition, emotion, and motor function [29,30]. Indeed, non-cognitive parameters in this task—such as immobility, thigmotaxis, and locomotor activity—serve as reliable indices of affective and motor phenotypes [29,30]. Immobility is interpreted as a depressive-like strategy akin to behavioral despair in the forced swim test, while excessive thigmotaxis reflects an anxiogenic-like response comparable to avoidance behavior in the elevated plus maze. Locomotor activity reflects motor performance and motivational drive, domains often assessed in the open field or rotarod.

In neuropsychiatric symptoms in the Morris water maze paradigm, scopolamine induced hyperlocomotion as indicated by higher values of speed velocity in probe trials, and augmentation of immobility and thigmotactic behavior in all sessions performed. Hyperlocomotion induced by scopolamine in behavioral tasks has been reported by several studies, including open field and Morris water maze tests [61,72,73]. Serotonin modulates the excitability threshold, which the blockade of pre-synaptic muscarinic receptors in the raphe nucleus reduces serotonin release in the hippocampus and septal areas, which results in hyperlocomotion [73,80,81]. Although controversial studies claim to explain the neurotransmitter influence of muscarinic blockade, dopamine tonus also was indirectly modified by muscarinic blockade, which the augmentation of dopaminergic transmission displays hyperlocomotion [72,73]. Our results showed that the treatment with 1N2PE synergically reinforced the hyperlocomotion, which may be explained by the blockade of serotonin and dopamine transporters, as well as the inhibitory activity on AChE, resulting in higher levels of serotonin, dopamine and acetylcholine in certain brain areas, and consequently hyperlocomotion, increasing the mean and maximum speed during the swimming task. Conversely to scopolamine-induced cognitive deficits, the 1N2PE-treated animals recover learning and memory efficiency, which demonstrates that hyperlocomotion induced by the bioactive molecule did not affect cognition.

Although scopolamine protocol elicited hyperlocomotion, immobility time was also seen in these animals in the 4th training and probe sessions. Despair behavior has been developed by the Porsolt protocol and exhibits some similarities with the Morris paradigm [82]. Thus, non-cognitive elements have been recorded, such as freezing and immobility time [29,30]. The muscarinic blockade induced the percentual of immobility time increase in the 4th trial and test session, which suggests depressive-like behavior [29]. Contradictory studies have postulated the effects of muscarinic blockers on depressive-like behavior assessed on swimming tasks [65,83]. We found that repetitive exposure to the pool induced a stressful response related to the augmentation of immobility in the scopolamine groups, but not in the 1N2PE-treated animals. Our theory relies on the pleiotropic actions of the 1N2PE, which may improve serotonergic and dopaminergic signaling, restoring the mobility status in the Morris paradigm [5,13].

Thigmotaxis consists of a natural behavior in open tasks that has been associated with anxiety-like status [84,85,86]. Thigmotactic behavior in the early learning trials predicts poor scores in recall place assessment since it affects behavioral strategy [84]. Such behavioral response reflects the level of emotional stress against an aversive environment, associated with higher levels of circulating corticosteroids [87,88]. Our data demonstrated that the scopolamine group adopted a thigmotactic phenotype, reducing the mean distance to the border in the 2nd trial of the first day of training and the probe trials. In fact, in the early training exposures, an augmentation of thigmotaxis is expected; however, higher levels of thigmotactic profile in the probe trail following 4 training days with 4 trials per day suggest anxiety-like behavior [29]. Studies have indicated the anxiety-like effects induced by scopolamine administration in the elevated plus maze and open field tests, which may be related to monoamine levels reduction in motivational brain structures [61,65,72,73]. The 1N2PE treatment improved the thigmotaxis behavior related to scopolamine and control groups, suggesting that the bioactive molecule seems to be still more effective on motivational non-cognitive tasks, especially anxiety-like phenotype. Such effectiveness is supported by the strong interaction of the 1N2PE with SERT and DAT, facilitating serotonergic and dopaminergic signaling, and mitigating anxiety-like responses [5].

The results of in silico and in vivo approaches suggest that 1N2PE consists of a molecule with multimodal characteristics. The deliberate and rational design of drugs that act on multiple targets has gained prominence in the last decade. The profile of multi-target drugs indicates molecules that produce additive or synergistic effects, while reducing side effects, contributing to therapy in refractory patients. Several neuropsychiatric diseases are multifactorial, involving several signaling pathways, complicating specific target modulation therapy.

In this context, the results highlight the potential of 1N2PE as a bioactive compound for the treatment of diseases of the central nervous system, especially cognitive and affective disorders. Recently, we published a literature review on 1N2PE and highlighted its significant neuropharmacological effects [13], which, combined with the present findings, allowed us to identify possible mechanisms of action and key molecular targets that deserve further investigation, despite the limitations of this study.

Limitations should be considered when interpreting the results of this study. First, although male rats were used as a model for scopolamine-induced cognitive impairment, extrapolation to other species, sexes, or humans should be done cautiously. Second, while the model allows evaluation of cognitive, emotional, and motor outcomes, it cannot fully capture the complexity of human neuropsychiatric conditions. Despite these limitations, the study provides insightful evidence of the effects of scopolamine and the potential modulatory role of 1-nitro-2-phenylethane.

## 4. Materials and Methods

### 4.1. Extraction and Isolation of 1-Nitro-2-Phenylethane

The plant material of *Aniba canelilla* (Kunth) Mez (Lauraceae) was collected in Ulianópolis, Pará, Brazil, and identified by comparison with an authentic exsiccate (MG 174904) deposited in the Herbarium João Murça Pires of the Emílio Goeldi Museum, Belém, Pará, Brazil. The trunk wood was dried, pulverized in a mill, and subjected to continuous hydrodistillation for 3 h using a modified Clevenger-type apparatus coupled to a refrigeration system maintained at 10 °C. The obtained essential oil was centrifuged, dried with anhydrous sodium sulfate, and stored at 5–10 °C until use. For the isolation of 1-nitro-2-phenylethane (1N2PE), 1.0 g of the essential oil was fractionated on a silica gel 60 chromatography column (70–230 mesh) and eluted with n-hexane/ethyl acetate mixtures in increasing polarity (0–15% ethyl acetate). The resulting fractions were concentrated on a rotary evaporator and analyzed by thin-layer chromatography (TLC) and gas chromatography with flame ionization detection (GC-FID) to identify and quantify 1N2PE, following previously established protocols [19,59].

### 4.2. In Silico Assessment of the Pharmacokinetics Profile

In silico evaluation for the 1N2PE molecule (Figure 7) was performed by SwissADME and ADMETlab 2.0, two online software tools provided by the Swiss Institute of Bioinformatics and the Computational Biology of Drug Design group at Central South University, to provide a global assessment of the pharmacokinetic profile and physicochemical and drug-associated information of the 1N2PE, respectively [89]. The prediction of physicochemical and pharmacokinetic parameters, such as water solubility, permeability across the cell membrane, P-glycoprotein inhibition potential, the volume of distribution, the elimination half-life, hepatic metabolism, and renal excretion were recorded [32].

### 4.3. Preparation of the Molecular Docking In Silico

The 1N2PE mechanism of action is still poorly understood. Thus, molecular docking was employed to predict the pharmacological pathways supporting the 1N2PE therapeutic probable effects on the principal pathophysiological systems targets involved in central nervous system disorders [90]. The Molegro Virtual Docker (MVD) program was selected for docking studies due to its docking algorithm called MolDock score, based on heuristic search, which combines differential evolution with a cavity prediction algorithm [91]. The MVD applies another reclassification scoring function to increase docking accuracy, identifying the most promising among MolDock results [92]. The 3D molecular structure of 1N2PE was drawn using Avogadro 1.2.0 software. Then, the 1N2PE structure was submitted to the MMFF94 type energy minimization calculation to eliminate errors that may occur in the structure’s design. The 1N2PE structure used in all computational procedures was obtained through quantum calculations performed with the Gaussian 09 software [93], employing Density Functional Theory (DFT) with the hybrid functional B3LYP. To ensure greater accuracy and efficiency in modeling atomic interactions, the 6-31G(d,p) basis set was adopted, which allowed the determination of the energetically most stable conformation. The 3D structures of the main targets of neuropsychiatric disorders were obtained from the Protein Data Bank (PDB) which includes acetylcholinesterase enzyme-AChE (7E3H), dopamine transporter-DAT (4M48), gamma-aminobutyric acid A-GABAA receptor (6X3X), prostaglandin-H synthase-PGHS-1 (1FE2), and serotonin transporter-SERT (5I6X). Subsequently, the pKa values were calculated using the ProPKA software, which predicts the pKa values of amino acids in the catalytic site, allowing the determination of protonation states under different pH conditions [94]. The MolDock Score function was used with a resolution of 0.3 Å and a cutting radius of 12 Å covering the entire catalytic cavity. Linker preparation was performed with internal ES, H-bond internal, and Sp2-Sp2 torsion parameters. The number of runs was 10, the maximum interactions were 1500, the maximum population size was 50, the maximum number of cycles was 300, and the maximum number of conformations was 5 for each system.

### 4.4. Preparation of Molecular Dynamics

The molecular dynamics simulation was performed by AMBER1449 using PMEMD accelerated by GPU. The strength fields AMBER ff14SB50 and TIP3P were used for proteins and water, respectively. The strength field parameters of the 1N2PE by the General AMBER Force Field were described [95]. Electrostatic potential calculations were performed using Gaussian09 at the level HF/6-31G*. For each simulation, a protocol involving energy minimization, heating, and equilibration was applied. Molecular dynamics simulations were performed for 100 ns under an NPT ensemble, with a temperature of 300 K and a pressure of 1 atm, using periodic boundary conditions. Long-range electrostatic interactions were calculated using the particle mesh Ewald method with a cutoff of 10.0 Å for direct spatial interactions. All bonds involving hydrogen atoms were constrained using the SHAKE algorithm, allowing an integration time step of 2 fs (femtoseconds) [96].

### 4.5. Binding Free Energy Calculation

The calculation of the binding free energy (MM/GBSA) of each system, excluding the entropic contribution, was computed using the trajectory generated by the MD simulation. The last 500 ps (picoseconds) were used for the calculation of free energy. The binding free energy was calculated using the formula:Δ𝐺biding = 𝐺𝑐𝑜𝑚𝑝𝑙𝑒𝑥 − [𝐺𝑟𝑒𝑐𝑒𝑝𝑡𝑜𝑟 + 𝐺𝑐𝑜𝑚𝑝𝑙𝑒𝑥].(2)

The binding free energy was also estimated by two additional terms, as observed in the formula:Δ𝐺biding = Δ𝐺𝑀𝑀 + Δ𝐺𝑠𝑜𝑙 − 𝑇Δ𝑆 = −𝑅.𝑇.ln [𝐾𝑒𝑞](3)

The Δ*G*MM represents the molecular mechanical energy and determines the sum of the intramolecular energy of the complex; ΔGsol represents the solvation energy, which consists of the sum of the polar contributions (ΔGPB/GB), and non-polar contributions (ΔGSA); TΔS consists of the conformational change in entropy associated with ligand binding to the receptor [97,98,99].

### 4.6. In Vivo Procedures

The 1N2PE was obtained from the trunk wood of *Aniba canelilla* (Kunth) Mez collected in the Amazon region and isolated as previously published by our group [19]. Animals’ procedure was approved by the Ethics Committee on Experimental Animals of the Federal University of Pará (UFPA)-Brazil, under number 2937220818, and followed the guidelines for the Care and Use of Laboratory Animals from the National Institutes of Health. Male Wistar rats (60 days old) were obtained from the UFPA animal facility, and maintained under controlled conditions (temperature, 12 h light-dark cycle, humidity) with food and water ad libitum. Briefly, animals were randomly divided into 3 groups (n = 8 animals/group). Randomization was organized using a systematic approach to ensure unbiased distribution and manage the experimental groups, with all animals being marked with different colors to facilitate group identification.

The protocol was divided into two phases. Firstly, the cholinergic muscarinic receptor antagonist scopolamine (1 mg/kg) or saline solution was administered by intraperitoneal (i.p.) route for 7 days (from 1st to 7th days of the protocol). In the second phase (from the 8th to 12th days of the protocol), the 1N2PE (5 mg/kg) or saline solution was administered i.p. for an additional 5 days, as previously validated by our study [19]. From the 8th to 12th days, animals were conducted to the controlled behavioral room (light 12 lux, temperature ±23 °C, attenuated noise) and submitted to the Morris water maze apparatus (circular pool 183 cm × 58 cm diameter vs height) with an acrylic platform submersed (2-cm below the surface) in the black dye (non-toxic soluble) liquid for the training session. The apparatus was virtually divided into 4 equal quadrants, and an additional subdivision of 20 cm in the outer circumference delimited the peripheric zone to evaluate thigmotaxis [29,30]. The training session consists of 4 consecutive days with 4 attempts per day with a 30-s inter-trial interval [21]. The training consisted of individually placing the animal in a distinct quadrant per session, allowing free swimming with a cut-off of 60 s to find the hidden platform (permanent place). In the failing task occurrence, the animal was gently forwarded to the platform. Fifteen seconds on the platform were permitted for all individuals to reach the target. On the 12th day of the protocol, animals performed the test day, in which the hidden platform was removed, and the animals were able to explore the tank for 60 s (Figure 8). To minimize potential bias during behavioral assessments, investigators were blinded to group allocation. Group assignment was maintained exclusively by the authors who did not participate in the behavioral evaluations.

Spatial learning and memory measurement was assessed through the latency to find the platform in the 4 sessions of the first day of training and distance to enter the platform zone during the probe parameters. The neuropsychiatric symptoms were evaluated through the swimming speed during the probe, thigmotactic behavior and percentage of immobility time during the 4-trails of the first day of the training and the test sessions [29,30]. Thigmotaxis was measured through the mean distance and the maximum distance from the border [29]. The percentage of immobility was assumed in the occurrence of a non-moving floating episode, which was evaluated according to the formula:(4)$$\%\;Immobility=\left(\frac{Immobility\;time}{\left(Immobility\;time+mobility\;time\right)}\right)\times 100$$

The 4 sessions of the first day of training were chosen for detailed measurements, since the early training motivational behavior has been related to the spatial performance on the probe trial [100]. In non-cognitive behavioral elements, swim speed is admitted as locomotor activity, thigmotactic behavior as anxiety or stress index, and immobility time as depressive-like behavior [29,30]. The behavioral test was recorded and analyzed by the video tracking ANY-maze™ software (Stoelting Co., Wood Dale, IL, USA).

### 4.7. Statistical Analysis

The Kolmogorov-Smirnov test was performed as a normality test. As the results indicated that the data did not follow a normal distribution and were non-parametric, one-way ANOVA followed by Tukey’s multiple comparisons or Kruskal–Wallis test corrected by Dunn’s multiple comparisons were applied according to Gaussian values. Data were expressed as mean ± SEM and significance level of *p* < 0.05 was adopted. GraphPad Prism 8.0 software was used for statistical analysis and graphical construction.

## 5. Conclusions

The present study demonstrates the potential of 1-nitro-2-phenylethene (1N2PE), a constituent of Aniba canelilla, as a multitarget candidate for the treatment of cognitive and psychiatric disorders. In silico analyses indicated pharmacokinetic properties compatible with central-acting drugs and relevant interactions with various molecular targets associated with cognition, mood, and neuroinflammation, including dopamine and serotonin transporters, acetylcholinesterase, and prostaglandin H synthase. Furthermore, the in silico results also demonstrated that the GABA system was not favorable. Complementarily, behavioral assays showed that 1N2PE treatment was able to reverse scopolamine-induced spatial memory deficits, as well as attenuate anxiety- and depression-related behaviors. These findings suggest that the molecule acts in an integrated manner on multiple neurotransmitter and inflammatory pathways, positioning it as a promising multitarget agent for the management of multifactorial neurological conditions. This study advances the field by combining computational and preclinical approaches, contributing to the initial pharmacological characterization of 1N2PE and expanding knowledge of its therapeutic potential. Future investigations, including biochemical and translational analyses, will be crucial to further elucidate its mechanisms of action and assess its applicability in more advanced studies, including clinical trials.

## Figures and Tables

**Figure 1 pharmaceuticals-18-01511-f001:**
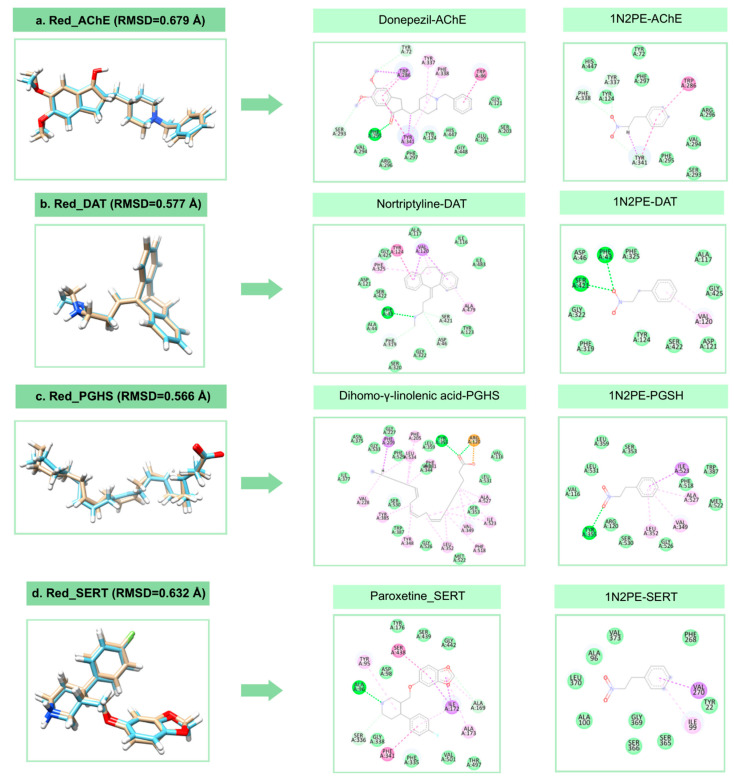
Intermolecular interactions of reference drugs with their molecular targets obtained by redocking: dopamine transporter (Red_DAT), serotonin transporter (Red_SERT), acetylcholinesterase (Red_AChE), prostaglandin-H synthase 1 (Red_PGHS), and the GABAA_AA receptor (Red_GABAA). Corresponding interactions of 1N2PE with the same targets are also shown: 1N2PE_DAT, 1N2PE_SERT, 1N2PE_AChE, 1N2PE_PGHS, and 1N2PE_GABAA.

**Figure 2 pharmaceuticals-18-01511-f002:**
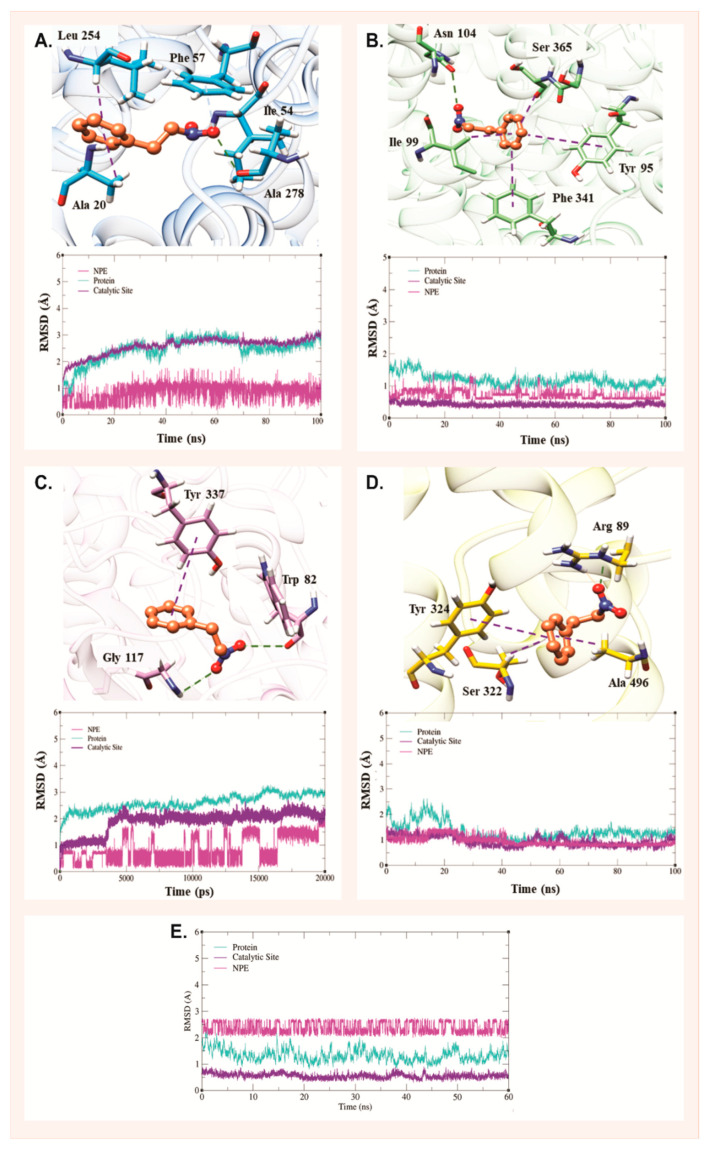
Interactions of 1-nitro-2-phenylethane (1N2PE) with potential molecular targets: (**A**) dopamine transporter (DAT), (**B**) serotonin transporter (SERT), (**C**) acetylcholinesterase (AChE), and (**D**) prostaglandin-H synthase 1 (PGHS-1). (**E**) Root mean square deviation (RMSD) of interactions with the γ-aminobutyric acid receptor (GABAA).

**Figure 3 pharmaceuticals-18-01511-f003:**
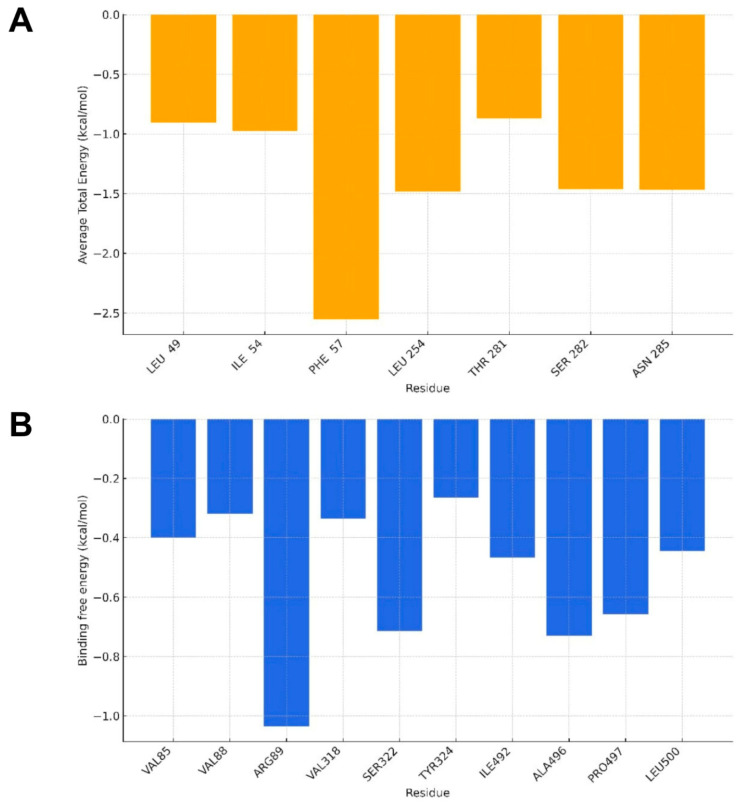
(**A**) Decomposition by residue of the DAT system, (**B**) decomposition by residue of the PGHS system.

**Figure 4 pharmaceuticals-18-01511-f004:**
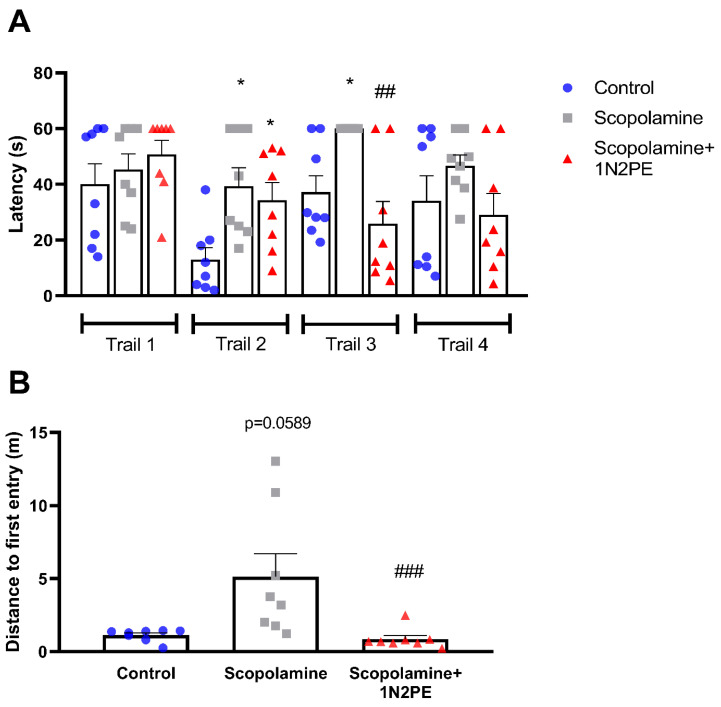
Effects of 1-nitro-2-phenylethane (1N2PE; 5 mg/kg; ip.) on the spatial learning and memory deficits induced by scopolamine in the Morris Water Maze paradigm (*n* = 8 animals/group). Escape latency in the 4 trials of the first day of training section (panel (**A**)) and distance to first entry in the target quadrant (panel (**B**)) were measured. Results are expressed as mean ± S.E.M. * *p* < 0.05 compared to the control group; ## *p* < 0.01 compared to the scopolamine group; ### *p* < 0.001 compared to the scopolamine group. One-way ANOVA followed Tukey’s test (latency to find the platform) and Kruskal–Wallis test followed Dunn’s multiple comparisons test (distance to first entry).

**Figure 5 pharmaceuticals-18-01511-f005:**
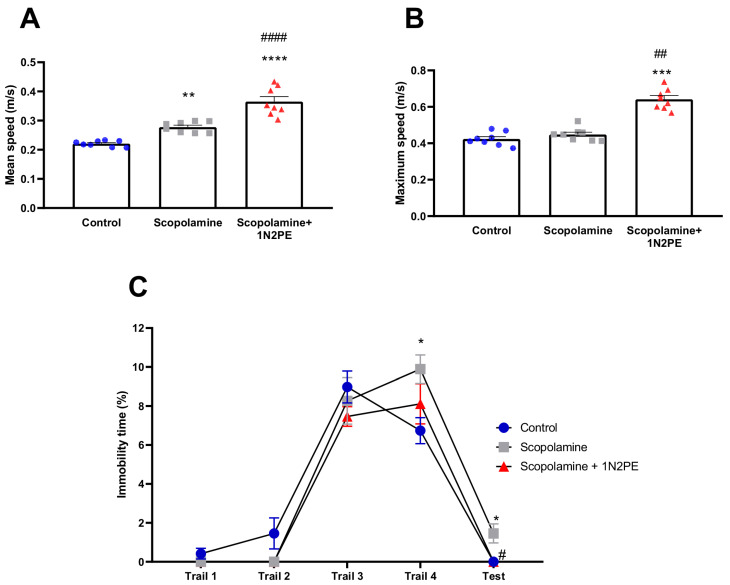
Effects of 1-nitro-2-phenylethane (1N2PE; 5 mg/kg; ip.) on the neuropsychiatric symptoms induced by scopolamine in the Morris Water Maze paradigm. Mean speed (panel (**A**)), maximum speed (panel (**B**)), and percentual of immobility time in the 4 trials of the first day of training section and probe trial (panel (**C**)) were measured (*n* = 8 animals/group). Results are expressed as mean ± S.E.M. * *p* < 0.05 compared to the control group; ** *p* < 0.01 compared to the control group; *** *p* < 0.0005 compared to the control group; **** *p* < 0.0001 compared to the control group; # *p* < 0.05, ## *p* < 0.01, #### *p* < 0.0001 compared to the scopolamine group. One-way ANOVA or Mixed-effects model (REML) followed by Tukey’s test (mean speed and % immobility, respectively) and Kruskal–Wallis test followed by Dunn’s multiple comparisons test (maximum speed).

**Figure 6 pharmaceuticals-18-01511-f006:**
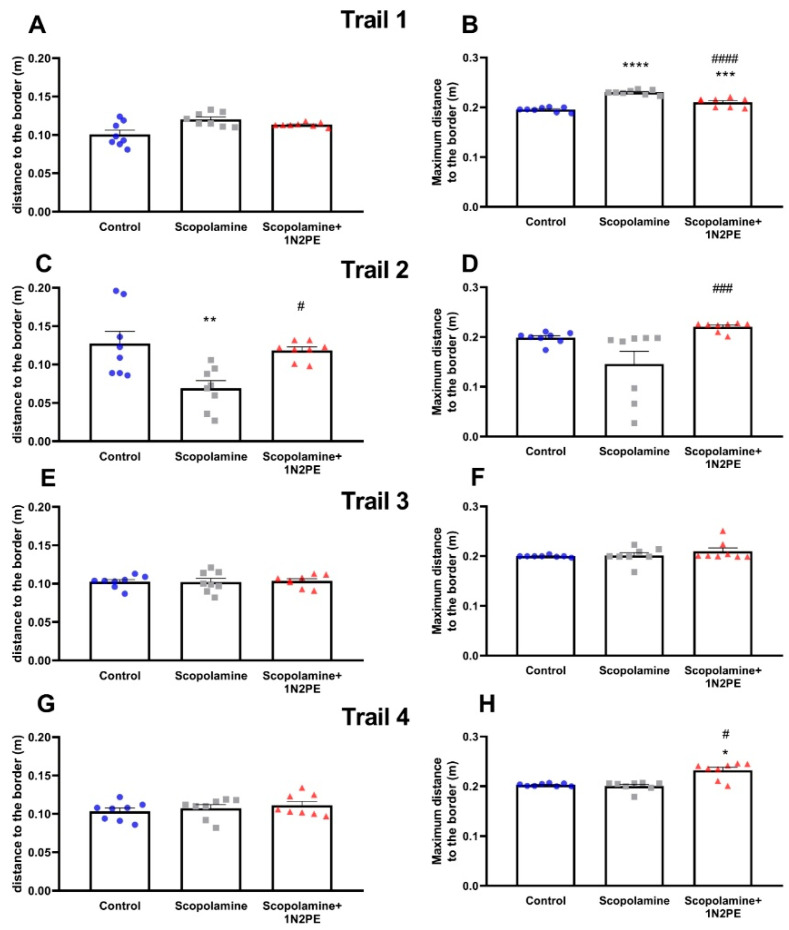
Effects of 1-nitro-2-phenylethane (1N2PE; 5 mg/kg; ip.) on the thigmotaxic neuropsychiatric symptoms induced by scopolamine in the Morris Water Maze paradigm. Mean distance to the border (panel (**A**,**C**,**E**,**G**,**I**)), and maximum distance to the border (panel (**B**,**D**,**F**,**H**,**J**)) in the 4 trails of the first day of training section and probe trial were measured (n = 8 animals/group). Panel (**K**) shows the representative tracking plot of the thigmotaxis behavior through the mean distance to the border of the 60-s probe session. Black circle represents the 20-cm thigmotaxis zone. Results are expressed as mean ± S.E.M. * *p* < 0.05, ** *p* < 0.01, *** *p* < 0.001, **** *p* < 0.0001 compared to the control group; # *p* < 0.05, ### *p* < 0.001, #### *p* < 0.0001 compared to the scopolamine group. One-way ANOVA followed by Tukey’s test or Kruskal–Wallis test followed by Dunn’s multiple comparisons test.

**Figure 7 pharmaceuticals-18-01511-f007:**
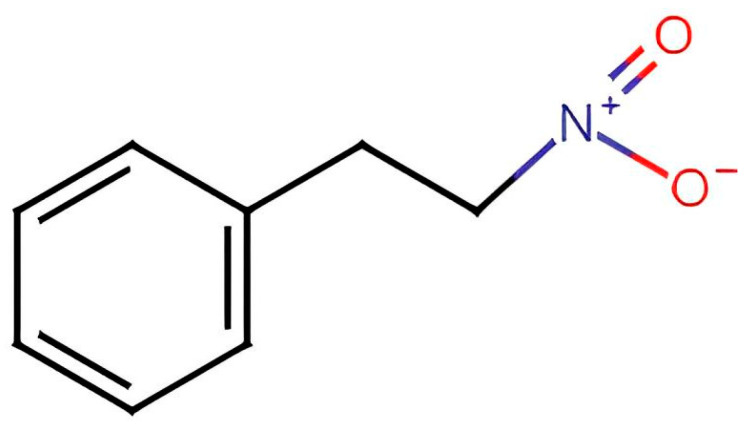
2D Molecular structure of 1-nitro-2-phenylethane.

**Figure 8 pharmaceuticals-18-01511-f008:**
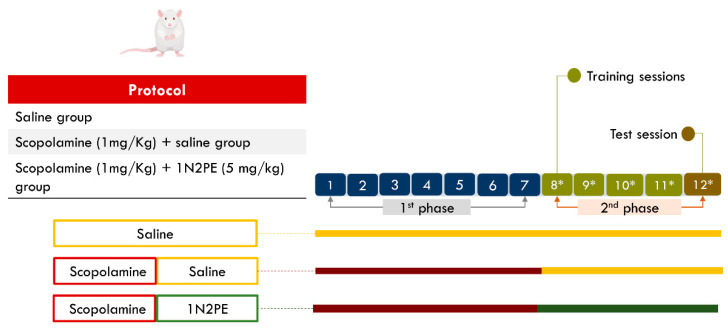
Schematic overview of the experimental protocol. In the first phase, scopolamine or saline was administered intraperitoneally (i.p.) for 7 consecutive days. In the second phase, 1-nitro-2-phenylethane (1N2PE; 5 mg/kg, i.p.) or saline was administered for 5 days. From the 8th day onward, animals were subjected to the Morris Water Maze paradigm, which consisted of locating a submerged platform during training sessions (days 8–11) and performing a platform-free probe test on day 12. ^∗^ Indicates the period in which behavioral testing was conducted.

**Table 1 pharmaceuticals-18-01511-t001:** Physicochemical parameters of 1N2PE calculated from the SwissADME software.

Physical-Chemical Parameters	Values	Pharmacokinetics	Result	Drug-Likeness	Result
Solubility (LogS)	−2.29	GI absorption	High	Lipinski	0 violation
Lipophilicity (LogP)	2.03	BBB permeant	Yes	Eagan	0 violation
pKa	9.21	P-gp substrate	No	Veber	0 violation
Molecular weight	151.16 g/mol	Skin permeation	−5.75 cm/s	Bioavailability Score **	0.55

** Determining a bioavailability score is necessary to calculate the probability that a target candidate has at least 10% oral bioavailability in mice, or a significant Caco-2 permeability is not less than 0.25.

**Table 2 pharmaceuticals-18-01511-t002:** Pharmacokinetics parameters of 1N2PE calculated from the ADMETlab software.

ADME	Absorption	Value	Distribution	Value	Metabolism	Value	Excretion	Value
Parameters	Caco-2 permeability	−4.27 Log unit	PPB	81.55%	CYP1A2	I/S	CL	8.77 mL/min/kg
	MDCK permeability	0.26 × 10^−4^ cm/s	FU	16.12%	CYP2C19 CYP2C9 CYP2D6 CYP3A4	I/S I/S I/S I/S	T1/2	45 min
	HIA	0.6 × 10^−2^ cm/s	VD	0.81 L/kg				

Abbreviations: HIA (Human intestinal absorption); PPB (Plasma Protein Binding); VD (Volume of Distribution); FU (Fraction unbound in plasma); CL (Clearance); T1/2 (half-life); I (inhibitor) and S (substrate).

**Table 3 pharmaceuticals-18-01511-t003:** Anchoring energy results obtained by the MolDock score.

Target	Binding Energy _(NPE)_ (kcal/mol)	DS_norm (NPE)_ (kcal/mol)	Binding Energy _(control)_ (kcal/mol)	DS_norm (control)_ (kcal/mol)
SERT	−68.00	−93.61	−108.28	−115.30
DAT	−62.53	−83.08	−107.09	−123.31
GABA_A_	−52.07	−71.68	−113.09	−124.69
AChE	−59.66	−82.13	−131.28	−130.56
PGHS-1	−58.45	−80.46	−126.12	−125.31

Abbreviations: serotonin transporter (SERT); dopamine transporter (DAT); gamma-aminobutyric acid receptor (GABAA); acetylcholinesterase enzyme (AChE); prostaglandin-H synthase 1 (PGHS-1).

**Table 4 pharmaceuticals-18-01511-t004:** Free Energy [ΔG]/Standard deviation [Å].

Molecular Targets	GBSA
SERT	−18.20/1.59
PGHS-1	−20.27/1.86
AChE	−16.58/2.38
DAT	−26.26/1.92

## Data Availability

Data is contained within the article and Appendix A.

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
