# Peer review of "1-Nitro-2-Phenylethane as a Multitarget Candidate for Cognitive and Psychiatric Disorders: Insights from In Silico and Behavioral Approaches"

_pharmaceuticals, 2025, doi:10.3390/ph18101511_

Round 1

Reviewer 1 Report

Comments and Suggestions for Authors

Neuroinflamation is an actual problem for healthcare system. In presented manuscript Fonseca et al. focused on the research of 1-nitro-2-phenylethane (1N2P). The complex of in silico and in vivo methods was applied during the study. The results confirm the potential of 1N2P in prevention of neuroinflamation. This paper may be interesting for a wide range of healthcarem professinals in drug designers, and the topic suits to the scope of the Pharmaceuticals. However, several issues should be solved.

Major comments

1) The introduction looks too brief. The adding of epidemiological data on the neuroinflamation and description of current treatment methods may be benefitial and clearfy the actuallity of current research.

2) The including of resolution values in the text of paper will increase the clearity of modelling, that was performed.

3) To be sure, that the applied computitional method is appraopriate for the analised molecular system, the re-docking should be performed, and the RMSD will reflect the suitability.

4) Also, the RMSD data should be added to all docking results.

5) Additionally, the resolutions of 4M48, 6X3X, 1FE2, and 5I6X are higher 2.5, that increases the risk of bias in the docking result [10.3390/ijms23116023]. Probably, another biological targets should be selected.

6) In my oppinion, the presentation of figures of docking data and showing the bounds between ligand and biological targets will increase the clearity and the correlations with the molecular dynamic.

7) Why did author deside to use the Kruskal–Wallis analysis as statistical method?

8) What was the purity of 1N2P substance in the in vivo experiment? How was the purity controlled? Was the impurities identified?

9) How was the process of randomisation orgonised?

10) Were the investigators blinded during the in vivo experiment? It should be described in materials and method section to decrease the risk of bias?

11) Were any animals, that dropped out the experimet? How were these data administrated?

12) The limitation section should be added in the discution, where the weeknesses of current reserch will highlited.

13) Also, the transposibility of the rat model on the human desease should be discussed.

14) The abstract should contined some digital information too make the srticle more interesting for the readers.

Minor comments

15) The quality of Figure 1 and 5 shoud be imroved

16) The latin name of species should written as foollows: Genus species AUTOR OF CLASSIFICATION - see the examples: 10.33380/2305-2066-2024-13-2-1677, 10.33380/2305-2066-2024-13-1-1719

Author Response

Reviewer #1

Neuroinflammation is an actual problem for healthcare system. In presented manuscript Fonseca et al. focused on the research of 1-nitro-2-phenylethane (1N2PE). The complex of in silico and in vivo methods was applied during the study. The results confirm the potential of 1N2PE in prevention of neuroinflammation. This paper may be interesting for a wide range of healthcare professionals in drug designers, and the topic suits to the scope of the Pharmaceuticals. However, several issues should be solved.

Response: We sincerely appreciate the detailed suggestions provided for our manuscript and have carefully considered point-to-point, as follows:

  1. The introduction looks too brief. The adding of epidemiological data on the neuroinflammation and description of current treatment methods may be beneficial and clarify the actuality of current research.

Response: We thank the reviewer for this suggestion. While a full discussion of inflammation in neuropsychiatric disorders goes beyond the scope of our study, we have incorporated additional information to improve contextualization and clarify the relevance of our research (lines 52-83).

  1. The including of resolution values in the text of paper will increase the clarity of modelling, that was performed.

Response: We thank you for this relevant issue. We included the resolution values in the Figure 1 and in the text.

  1. To be sure, that the applied computational method is appropriate for the analyzed molecular system, the re-docking should be performed, and the RMSD will reflect the suitability.

Response: We thank the reviewer for this suggestion. We have incorporated the molecular redocking through Figure 1, which also presents the intermolecular interactions of all the reference systems studied, as follow: “Although crystallographic structures with resolutions below 2.5 Å are generally preferred for molecular docking, such high-resolution entries are not always available for all biological targets [24]. The structures selected for this study contained ligands co-crystallized in their active sites, which enabled redocking and validation of the docking protocol to ensure the reliability of predicted ligand–receptor interactions. Our redocking analyses yielded root mean squared deviation (RMSD) values below 2.5 Å for all targets (AChE: 0.679 Å, DAT: 0.577 Å, PGHS: 0.566 Å, SERT: 0.632 Å, GABA: 0.688 Å), confirming that the predicted poses closely matched the experimental conformations. Therefore, despite the slightly higher resolution of some structures, they are suitable for docking simulations and provide biologically meaningful insights into the binding sites (Figure 1). Figure 1 also presents the structures of the reference drugs in complex with their respective receptors, along with the RMSD values from redocking, highlighting the key intermolecular interactions within each active site. Notably, the docking of 1N2PE revealed that, although structurally distinct from the reference drugs, it establishes interactions with the same critical amino acid residues at the receptor binding sites” (lines 177-191).

  1. Also, the RMSD data should be added to all docking results.

Response: We thank the reviewer for this suggestion. We have included the RMSD data in all docking results, as presented in lines 181-191.

  1. Additionally, the resolutions of 4M48, 6X3X, 1FE2, and 5I6X are higher 2.5, that increases the risk of bias in the docking result [10.3390/ijms23116023]. Probably, another biological targets should be selected.

Response: We thank you for your valuable suggestion. The targets were selected based on their well-established roles in the pathophysiology of several neurological and psychiatric disorders. As suggested, we included the information about the resolution values in the text, as follow: “Although crystallographic structures with resolutions below 2.5 Å are generally preferred for molecular docking studies, not all selected targets have PDBs available within this criterion. The chosen structures also contain ligands crystallized in the active site, allowing redocking and protocol validation to ensure the reliability of the predicted ligand–receptor interactions. Furthermore, the redocking performed in this study resulted in RMSD values below 2.5 Å, demonstrating that the obtained poses are consistent with the experimental conformations. Therefore, despite the slightly higher resolution, the selected structures are suitable for docking simulations and provide biologically relevant information about the binding sites” (lines 177-187).

  1. In my opinion, the presentation of figures of docking data and showing the bounds between ligand and biological targets will increase the clarity and the correlations with the molecular dynamic.

Response: Your feedback is very important to us. We have added Figure 1 showing the intermolecular interactions of 1N2PE with the biological targets and established the correlation with molecular dynamics, as observed in lines 178-191 and 500-511.

  1. Why did the author decide to use the Kruskal–Wallis analysis as a statistical method?

Response: This is an important question. Before we selected the statistical test, we conducted the Kolmogorov–Smirnov test to assess the normality of our data distribution. As the results indicated that the data did not follow a normal distribution and were non-parametric, we chose the Kruskal–Wallis test to appropriately compare differences among the groups. To improve the understanding of our research, we have added this information in the manuscript (lines 874-875).

  1. What was the purity of 1N2PE substance in the in vivo experiment? How was the purity controlled? Were the impurities identified?

Response: Thank you for pointing this out, it is important to clarify this information. In the present work, we used the isolated substance with a 100% purity of 1-nitro-2-phenylethane (1N2PE) as previously described by our group in de Campos et al. (2023; doi: 10.1016/j.jep.2022.116036). Therefore, to clarify this important aspect of our research we have added this information to the manuscript as follows: “As previously described by our group in de Campos et al. (2023), the plant material of Aniba canelilla (kunth) Mez (Lauraceae) was collected in Ulianópolis, Pará, Brazil, and identified by comparison with authentic exsiccate (MG 174904) maintained in the Herbarium João Murça Pires, from the Emílio Goeldi Museum, in the Belém city, Pará state, Brazil. The trunk wood previously dried for some days and pulverized in a mill was used to obtain the essential oil. The oil was hydrodistilled continuously (3h) using a modified Clevenger-type apparatus coupled to a refrigeration system at 10 °C. The oil was centrifuged and dried with anhydrous sodium sulfate and then maintained in a refrigerator at 5–10 °C for further assays” (lines 733-746).

Our research group has been dedicated to investigating the 1N2PE, including its antioxidant properties and its impact on behavioral impairments in animal models. In previous studies, we used the A. canelilla essential oil with high purity levels for 1N2PE (over 75%) and obtained similar results with prominent findings for its potential for treating diseases associated with inflammation and oxidative stress (doi: 10.3390/antiox11101903; doi: 10.1016/j.jep.2022.116036). However, studies with 1N2PE are still scarce, underscoring the need for further investigations to fully characterize its pharmacological profile and potential therapeutic applications, highlighting the importance of the present investigation.

  1. How was the process of randomization organized?

Response: Upon arrival from the Federal University of Pará (Universidade Federal do Pará, in portuguese) central vivarium, animals underwent a 5-day habituation period to hands-on management. Afterwards, they were then randomly assigned into three experimental groups, with 8 animals per group. To further clarify this topic we have added this information to the manuscript as follows: “Randomization was organized using a systematic approach to ensure unbiased distribution and manage the experimental groups, with all animals being marked with different colors to facilitate group identification” (lines 826-828).

  1. Were the investigators blinded during the in vivo experiment? It should be described in materials and method section to decrease the risk of bias?

Response: This is indeed a critical point in behavioral evaluations, thank you for this question. Group allocation was blinded to the investigators responsible for conducting the behavioral assessments (FJCS-Junior, BCC, and CACL) in order to minimize potential bias. The information regarding group assignment was kept exclusively by the first three authors (ECMF, LVPSP, and DLC), who were not involved in the behavioral evaluations. This procedure was implemented to ensure that outcome assessments were conducted independently of knowledge about treatment conditions. We acknowledge your precise recommendation and have incorporated this information to our materials and methods section in the manuscript (lines 847-850).

  1. Were any animals, that dropped out the experiment? How were these data administrated?

Response: This is a great point. No animals were excluded from the study, and all collected data were included in the analyses.

  1. The limitation section should be added in the discussion, where the weaknesses of current research will be highlighted.

Response: Thank you for this suggestion, it is indeed important to identify the weaknesses of our research since they represent a fundamental point of improvement for future studies. Therefore, we have added this topic in the manuscript (lines 724-730).

  1. Also, the transposibility of the rat model on the human disease should be discussed.

Response: We appreciate this suggestion and acknowledge that our discussion lacked a solid explanation about the transposability of the scopolamine-induced rat model to human disease. We have now expanded the discussion to clarify this issue as follows: “Although the translational validity of any preclinical model has inherent limitations, the scopolamine paradigm is widely recognized for reproducing key aspects of the cholinergic dysfunction disorders, as observed in Alzheimer’s disease [62]. Scopolamine, a non-selective muscarinic receptor antagonist, produces transient impairments in hippocampal cholinergic signaling that result in spatial learning and memory deficits [63]. While the model has been traditionally validated for cognitive dysfunction, our results demonstrate that scopolamine also alters non-cognitive parameters, such as immobility and thigmotaxis, highlighting its potential to capture neuropsychiatric-like symptoms. In this sense, our findings shed new light on how this classical model can be interpreted beyond memory impairment. Consistent with its use as an Alzheimer-like paradigm, we observed that scopolamine not only elicited cognitive deficits but also reproduced behavioral alterations resembling psychiatric manifestations frequently reported in patients with Alzheimer’s disease [29,30].Clinically, psychiatric symptoms often emerge in the early stages of Alzheimer’s disease, whereas cognitive decline becomes more prominent in the mild to later stages [64]. Therefore, the scopolamine model provides a valuable framework to investigate both domains, enabling an integrated preclinical evaluation of cognitive and neuropsychiatric alterations (lines 594-610).

  1. The abstract should contain some digital information to make the article more interesting for the readers.

Response: Thank you for this suggestion, it is indeed important to make our abstract capture the reader’s attention. Therefore, we have added a few data in the abstract (lines 27-47).

  1. The quality of Figure 1 and 5 should be improved.

Response: We apologize for the low quality of Figures 1 and 5 in the original version. In the revised manuscript, these figures have been replaced with higher-resolution images to improve clarity and ensure better visualization.

  1. The latin name of species should written as follows: Genus species AUTOR OF CLASSIFICATION-see the examples: 10.33380/2305-2066-2024-13-2-1677,10.33380/2305-2066-2024-13-1-1719.

Response: Thank you for this observation. We have made the alterations in the manuscript (lines 91-95).

Reviewer 2 Report

Comments and Suggestions for Authors

The research article under the title “1-Nitro-2-phenylethane as a Multitarget Candidate for Cognitive and Psychiatric Disorders: Insights from In Silico and Behavioral Approaches” presents comprehensive evaluation of the interactions formed between 1N2PE and proteins that are important for different disorders. These results are supplemented by the examination of the bihavioral effects. The authors provided plausible explanations, supported by a plethora of bioinformatics tools. Based on this, I recommend that authors answer the following questions and clarify some of the points. After the revision, the manuscript will be suitable for publication in Pharmaceuticals.

The authors should change the following:

  1. The authors should present evidence of the biological activity of 1N2PE, as only its mixtures with other compounds in essential oils were described in the introduction.
  2. What are the amounts of 1N2PE in these oils?
  3. The authors should explain how the optimal structure of the compound was selected for the in silico investigations.
  4. The authors should compare the results of the ADMET analysis with similar compounds that are already in use, as these predictions often come with some limitations.
  5. The authors have mentioned that the molecular docking simulations were successful but they did not explain the binding positions of the compound and their biological significances
  6. The structure of 1N2PE should be given, as it is important for the binding affinity to observe specific interactions that exist in the bonded structures
  7. The authors should explain if the binding energies were sufficient to give stable conformers under the physiological conditions
  8. “results above 5.15 are from..” what are the units of this number?
  9. “strong affinity with the SERT” – which criteria was applied for this statement?
  10. The discussion should be supplemented with the exact interactions that are formed between compound and protein, as it is important to elucidate which of the structural moieties are responsible for the formation of these interactions.
  11. The authors should compare the MWM results with the previous studies on similar compounds.

Author Response

Reviewer #2

The research article under the title “1-Nitro-2-phenylethane as a Multitarget Candidate for Cognitive and Psychiatric Disorders: Insights from In Silico and Behavioral Approaches” presents comprehensive evaluation of the interactions formed between 1N2PE and proteins that are important for different disorders. These results are supplemented by the examination of the behavioral effects. The authors provided plausible explanations, supported by a plethora of bioinformatics tools. Based on this, I recommend that authors answer the following questions and clarify some of the points. After the revision, the manuscript will be suitable for publication in Pharmaceuticals.

Dear reviewer,

We would like to express our sincere gratitude for your willingness to evaluate this review and to recognize its scientific relevance. We truly appreciate your evaluation.

  1. The authors should present evidence of the biological activity of 1N2PE, as only its mixtures with other compounds in essential oils were described in the introduction.

Response: We thank the reviewer for this valuable comment. We agree that it is important to highlight the biological activity of 1-nitro-2-phenylethane (1N2PE). Several studies have reported direct evidence of 1N2PE activity. For instance, de Lima et al. (2009; doi:10.1016/j.phymed.2008.10.007) demonstrated its antinociceptive effect in animal models, while de Siqueira et al. (2010; doi: 10.1016/j.ejphar.2010.03.060) showed that 1N2PE induces a vago-vagal bradycardic and depressor reflex in normotensive rats, evidencing its cardiovascular activity. Antioxidant effects of 1N2PE were also described in radical scavenging assays (da Silva et al., 2007; doi: 10.1021/jf071928e), as well as in vivo experiments involving inflammatory pathways (Cardoso et al., 2022; doi: 10.3390/antiox11101903). In addition, enriched preparations containing over 99% 1N2PE reversed memory impairment in rodents through mechanisms associated with cholinergic modulation (de Campos et al., 2023; doi: 10.1016/j.jep.2022.116036). Behavioral studies further support its neuropharmacological relevance, since 1N2PE reduced anxiety-like behavior in the elevated plus maze and exhibited anticonvulsant properties mediated by GABAergic pathways (Oyemitan et al., 2013; doi: 10.1016/j.phymed.2013.07.005). Together, these findings demonstrate that 1N2PE is an active compound with antinociceptive, cardiovascular, antioxidant, and neurobehavioral properties. In the revised version of the manuscript, we now clarify this distinction more explicitly and include these direct reports to strengthen the evidence that 1N2PE itself displays relevant biological activities, while also recognizing that further studies are required to explore its mechanisms in models of neuroinflammation and oxidative stress (lines 103-114). Recently, we also published a review article that compiles the main pharmacological properties of 1N2PE, with emphasis on its role in inflammatory-oxidative pathways (Pantoja et al., 2025, doi: 10.3389/fphar.2025.1552295).

  1.  What are the amounts of 1N2PE in these oils?

Response: Thank you for this question, we have added this information in the manuscript (lines 733-746). As also clarified ‘in response to Reviewer 1, in the present study we used isolated 1-nitro-2-phenylethane (1N2PE) at 100% purity. Regarding the essential oil, our previous chemical analyses of Aniba canelilla (Kunth) Mez (Lauraceae) demonstrated that 1N2PE is its major constituent, with levels above 75% of the oil composition (doi: 10.3390/antiox11101903; doi: 10.1016/j.jep.2022.116036). Nevertheless, detailed quantification may vary depending on the source and extraction conditions, which underscores the importance of studies employing the isolated molecule, as performed in the present work.

  1. The authors should explain how the optimal structure of the compound was selected for the in silico investigations.

Response: We thank you for the suggestion. We have included in the text details on how the atomic structure of the ligand was optimized, as well as the methods employed in this process, as follow: “The 1N2PE structure used in all computational procedures was obtained through quantum calculations performed with the Gaussian 09 software (FRISCH, M.; TRUCKS, G. W.; SCHLEGEL, H. B.; SCUSERIA, G. E.; ROBB, M. A.; CHEESEMAN, J. R.; et al. Gaussian 09 Revision D. 01. 2014.), employing Density Functional Theory (DFT) with the hybrid functional B3LYP. To ensure greater accuracy and efficiency in modeling atomic interactions, the 6-31G(d,p) basis set was adopted, which allowed the determination of the energetically most stable conformation” (lines 772-777).

  1. The authors should compare the results of the ADMET analysis with similar compounds that are already in use, as these predictions often come with some limitations.

Response: What an interesting suggestion. It is not possible to compare the ADME results of 1N2PE with therapeutically used similar compounds, as no analogous chemical structures are available at this moment. This is the robust value of this research. 

  1. The authors have mentioned that the molecular docking simulations were successful but they did not explain the binding positions of the compound and their biological significances.

Response: We thank you for the comment. To facilitate understanding, we have added Figure X, which shows that the docking of the 1N2PE ligand, although its structure is completely different from the reference drugs, still establishes interactions with the residues of the receptors’ active sites. These results suggest that the compound has affinity for the binding site. The potential biological activity was demonstrated in our rat model. This issue was broadly discussed in the discussion section. To clarify this issue, we included in the discussion section the sentence “We performed a redocking with reference drugs in the binding sites to validate the methods of docking.  In this context, the 1N2PE exhibited potential interactions with SERT, DAT, and AChE, and lower energy binding values for PGHS-1 and GABAA sites. Although the 1N2PE structure is completely different from the reference drugs, it still establishes interactions with the residues of the targets. These findings suggest the preferential site of action of the natural compound for serotonergic, dopaminergic, and cholinergic pathways in the affective and cognitive disorder theory. In this context, molecular docking was used to provide the most probable initial conformation of the 1N2PE ligand in the active site of the receptors. However, as a static and fast approach, its main utility lies in supporting molecular dynamics, which allows the evaluation of temporal stability, system flexibility, and conformational changes throughout the simulation” (lines 500-511).

  1. The structure of 1N2PE should be given, as it is important for the binding affinity to observe specific interactions that exist in the bonded structures.

Response: Thank you for this suggestion. We recognize that it is important to offer the structure of the compound. As recommended, we included Figures 3, which presents the residue decomposition and highlights the most relevant amino acids in the binding site. The main interactions are detailed in results and discussion section (lines 560-589).

  1. The authors should explain if the binding energies were sufficient to give stable conformers under the physiological conditions.

Response: We appreciate your suggestion for this analysis. The only system that did not behave appropriately under physiological conditions was the GABAA receptor, where the ligand was expelled from the binding site due to its high free energy, as described in lines 502-503. The other systems exhibited energetically favorable behavior under physiological conditions, since our molecular dynamics methods mimics physiological conditions, with negative binding free energy values, as shown in lines 552-559.

  1. “results above 5.15 are from..”what are the units of this number?

Response: We apologize for this mistake. The correct unit is “cm/s” and it refers to the skin permeation. We have added this information in the revised manuscript (line 440).

  1. “strong affinity with the SERT”–which criteria was applied for this statement?

Response: Thank you for the question. We apologize for the mistake and have already made the necessary correction in the paragraph. We now relate the stability observed in the RMSD results to the possibility of chemical affinity with the receptor, as follow: “Notably, the RMSD of the catalytic site remains stable in the presence of the ligand, suggesting the existence of a chemical affinity with the receptor” (lines 530-531 and 242-244).

  1. The discussion should be supplemented with the exact interactions that are formed between compound and protein, as it is important to elucidate which of the structural moieties are responsible for the formation of these interactions.

Response: We appreciate your suggestion. With the construction of Figure 3, we conducted a detailed discussion of the specific interactions formed between the compound and the protein, highlighting which structural moieties are responsible for these interactions and, consequently, for the stabilization of the system, as follow: “The residue-based free energy decomposition provides detailed insights into the individual contributions of amino acids to the stabilization of the ligand within their respective binding sites. Among the evaluated systems, the dopamine transporter (DAT) exhibited the lowest total free energy (-26.26 kcal/mol), followed by cyclooxygenase (PGHS-1) with -20.27 kcal/mol, as represented in Figures Ya and Yb, respectively.

In the DAT system, several residues displayed strong stabilizing contributions, which account for the more negative global free energy value. Notably, PHE57 established the most significant interaction, of the π–benzene ring type, with a contribution of -2.5 kcal/mol. This interaction plays a central role in anchoring the ligand to the binding site, reinforcing the structural complementarity between ligand and protein. In addition, other residues, although contributing with lower energetic values, also cooperated in the stabilization of the complex, such as LEU254, ILE54, SER282, and ASN285. These findings highlight that, in the DAT system, the predominance of stabilizing interactions not only confirms the higher global affinity of the ligand but also identifies critical residues that may be exploited in structural optimization strategies.

In the PGHS-1 system, the residue decomposition profile revealed weaker interactions compared to the DAT system, resulting in a less negative global free energy. The most prominent residue was ARG89, which established the strongest hydrogen bonds interaction, contributing -1.5 kcal/mol. Additionally, residues such as ALA496, SER322, and PRO497 exhibited more modest but still relevant contributions to the stabilization of the complex. The presence of these stabilizing interactions, although of lower magnitude compared to the DAT system, indicates that the ligand still retains affinity for PGHS-1, albeit with reduced steric and electronic complementarity.

Taken together, these results demonstrate that the ligand forms more favorable and consistent interactions within the DAT binding site, making it the most promising target in terms of complex stability. In contrast, while the ligand also interacts stably with PGHS-1, the lower intensity and fewer number of key interactions result in reduced affinity. Thus, the residue decomposition analysis not only corroborates the global free energy findings but also identifies key residues that may guide future structural modifications aimed at enhancing selectivity and improving the interaction profile across different molecular targets” (lines 560-589).

  1. The authors should compare the MWM results with the previous studies on similar compounds.

Response: Thank you for this suggestion. Traditionally, the Morris water maze (MWM) has been employed as a hippocampal-dependent task to evaluate spatial learning and memory (Morris, 1984; 10.1016/0165-0270(84)90007-4). In this sense, several studies have shown that essential oil derivatives exert beneficial effects in the MWM paradigm. For example, linalool reversed neuropathological impairments in Alzheimer’s disease models (Sabogal-Guáqueta, Osorio and Cardona-Gómez, 2016; 10.1016/j.neuropharm.2015.11.002). More broadly, essential oil-derived compounds have consistently improved MWM performance across diverse rodent models, reinforcing the translational relevance of small essential oils molecules, such as linalool and linalool-rich oils which improved escape latency and target quadrant occupancy in Alzheimer and Aβ models, effects linked to reduced neuroinflammation and oxidative stress (Sabogal-Guáqueta, Osorio and Cardona-Gómez, 2016, doi: 10.1016/j.neuropharm.2015.11.002; Yuan et al., 2021, doi: 10.1155/2021/8887716). Several other molecules have demonstrated improvements in cognitive functions in pharmacologically induced cognitive deficit models (e.g.: scopolamine (Kanojia et al., 2021, doi: 10.2174/1871527320666210202121103; Bigdeli, Asle-Rousta and Rahnema, 2019, doi: 10.1007/s11062-019-09800-0; Mehrjerdi et al., 2020, doi: 10.1007/s00210-020-01866-6). These findings show that small molecules from essential oils improve MWM measures through anti-inflammatory, antioxidant, and neuromodulatory mechanisms, as suggested by the neuropharmacological properties of 1N2PE (Pantoja et al., 2025, doi: 10.3389/fphar.2025.1552295).

Actually, our previous work also demonstrated that 1N2PE can mitigate memory impairments, highlighting its relevance for cognitive domains (de Campos et al., 2023; doi: 10.1016/j.jep.2022.116036). Given the central role of the cholinergic system in memory, we applied the scopolamine challenge to assess cognitive dysfunction (Alber et al., 2020, doi: 10.1186/s13195-020-00599-1; Aksoz, Akyol and Korkut, 2024, doi: 10.1016/j.bbr.2024.114978). Beyond cognition, scopolamine-induced cholinergic disruption is associated with neuropsychiatric manifestations (Murayama et al., 2021, doi: 10.1538/expanim.21-0009), thereby modeling both cognitive and psychiatric features. In this context, thigmotaxic behavior is particularly informative: excessive thigmotaxis hampers memory acquisition in the MWM (Higaki et al., 2018, doi: 10.1371/journal.pone.0197003; Murayama et al., 2021, doi: 10.1538/expanim.21-0009). Our data corroborate this, as scopolamine-challenged rats exhibited greater thigmotaxis and poorer memory performance compared with controls and 1N2PE-treated animals. In addition, scopolamine increased immobility and thigmotaxis relative to other groups, further validating these non-cognitive measures.

Although underutilized for non-cognitive purposes, the MWM thus offers a unique opportunity to capture neuropsychiatric-like dimensions in a single paradigm, providing an integrated view of cognition, emotion, and motor function (Murayama et al., 2021, doi: 10.1538/expanim.21-0009). Indeed, non-cognitive parameters in the MWM – such as immobility, thigmotaxis, and locomotor activity – serve as reliable indices of affective and motor phenotypes (Murayama et al., 2021, doi: 10.1538/expanim.21-0009). Immobility is interpreted as a depressive-like strategy akin to behavioral despair in the forced swim test, while excessive thigmotaxis reflects an anxiogenic-like response comparable to avoidance behavior in the elevated plus maze. Locomotor activity reflects motor performance and motivational drive, domains often assessed in the open field or rotarod.

This interpretation aligns with previous studies of natural products. Essential oils rich in linalool, such as those from Aniba rosaeodora, showed anxiolytic and neuroprotective effects in our group’s work, reducing immobility in the forced swim test, increasing open-arm exploration in the elevated plus maze, and protecting against oxidative stress (dos Santos et al., 2018, doi: 10.1016/j.jep.2017.10.013; dos Santos et al., 2022, doi: 10.2174/1570159X19666210920094504; dos Santos et al., 2024, doi: 10.1016/j.biopha.2024.117120). To improve the understanding of our research, we have added this information in the manuscript (lines 639-662).

We thank again the Editor and the Reviewers for their comments and helpful suggestions, which contributed to allow us to improve the manuscript. We hope that, by addressing all the comments of the Reviewers, the revised version of the manuscript may prove acceptable for publication in Pharmaceuticals.

Sincerely,

Cristiane Maia and Jofre J. S. Freitas

Corresponding authors.

Round 2

Reviewer 1 Report

Comments and Suggestions for Authors

The authors have performed an excellent job and the current manuscript is suitable for publication.

Reviewer 2 Report

Comments and Suggestions for Authors

The authors have responded to all of the comments correctly. The manuscript is suitable for publication in the present form.